# Context-Aware Meta-Learning

**Christopher Fifty**[1], **Dennis Duan**[1,2], **Ronald G. Junkins**[1],
**Ehsan Amid**[3], **Jure Leskovec**[1], **Christopher Ré**[1], **Sebastian Thrun**[1]
[1]Stanford University, [2]Google, [3]Google DeepMind
`fifty@cs.stanford.com`

## Abstract

Large Language Models like ChatGPT demonstrate a remarkable capacity to learn new concepts during inference without any fine-tuning. However, visual models trained to detect new objects during inference have been unable to replicate this ability, and instead either perform poorly or require meta-training and/or fine-tuning on similar objects. In this work, we propose a meta-learning algorithm that emulates Large Language Models by learning new visual concepts during inference without fine-tuning. Our approach leverages a frozen pre-trained feature extractor, and analogous to in-context learning, recasts visual meta-learning as sequence modeling over datapoints with known labels and a test datapoint with an unknown label. On 8 out of 11 few-shot image classification benchmarks, our approach—without meta-training or fine-tuning—exceeds or matches the state-of-the-art algorithm, P>M>F, which is meta-trained on these benchmarks. Our code is available at https://github.com/cfifty/CAML.

## 1 Introduction

Meta-learning refers to a capacity to learn new concepts from a small number of demonstrations (Lake et al., 2015). In a decade of remarkable advances to machine intelligence, it remains an area where human performance continues to surpass that of machines (Brown et al., 2020). To match human capabilities, and towards developing machines that can learn and think like humans, we must develop machine intelligence capable of learning novel concepts from only a few examples (Lake et al., 2017).

Many applications of deep learning apply a learning algorithm to a large set of training data; however, learning from a very small number of training examples poses a challenge (Lake et al., 2017; Garnelo et al., 2018). This challenge led to two predominant evaluation settings: *in-domain* and *cross-domain*. The *in-domain* setting evaluates a meta-learner's ability to quickly adapt to new tasks after training on similar tasks within a specific domain. Models designed for this setting are often extremely fast but exhibit poor generalization to tasks outside the target domain (Chen et al., 2019). Meanwhile, the *cross-domain* setting evaluates a meta-learner's ability to adapt to tasks in previously unseen domains. Methods designed for this setting are highly adaptable but slow during inference as they require fine-tuning on the support set (Guo et al., 2020; Oh et al., 2022; Hu et al., 2022). Critically, meta-learners in both settings differ from a human's capacity to quickly generalize to new tasks.

The problem of simultaneously fast and general meta-learning has recently been addressed in Natural Language by Large Language Models (LLMs). LLMs like ChatGPT can quickly generalize to new tasks through an ability termed in-context learning (Brown et al., 2020). However, it remains an open problem in Computer Vision. Even the best visual meta-learning algorithms cannot be deployed to a ChatGPT-like system because such systems require models that can (1) generalize to a broad set of tasks unknown at training time and (2) do so in real-time, without the time allowance for finetuning the model. LLMs have shown a remarkable ability to do both; however, current visual meta-learners may only satisfy one requirement or the other (Hu et al., 2022).

To measure progress towards this goal of fast and general visual meta-learners, we develop an evaluation paradigm that we call *universal meta-learning*. *Universal meta-learning* measures a model's capacity to quickly learn new image classes. It evaluates models across a diverse set of meta-learning benchmarks spanning many different image classification tasks without meta-training on any of the benchmarks' training sets or fine-tuning on the support set during inference. We focus on

the application of few-shot image classification—as opposed to dense prediction tasks like in-painting or segmentation—as the universal setting has already been explored for these applications (Bar et al., 2022; Zhang et al., 2023; Wang et al., 2023; Kim et al., 2023; Butoi et al., 2023).

Beyond benchmarking methods in the universal setting, we present a meta-learner that achieves strong universal performance. Drawing inspiration from in-context learning in LLMs, we reformulate $n$-way-$k$-shot image classification as non-causal sequence modeling over the support set and an unknown query image. Specifically, given $n$-way classification with $k$-examples from each class, we train a non-causal model over $\{(x_i, y_i)\}_{i=1}^{nk}$ (image, label) support set pairs, and an unlabeled query image $x_{nk+1}$, to predict the label of the query image. This formulation causes the meta-learner to extrapolate to new classes in its parameter space, enabling it to learn new visual concepts during inference without fine-tuning. Due to its capacity to learn visual information "in-context", we term our approach *Context-Aware Meta-Learning* (CAML).

In summary, our contribution is two-fold. First, we develop a meta-learning evaluation paradigm that approximates the performance of visual meta-learners in a ChatGPT-like application. Second, we design a meta-learning algorithm that works well in this setting. Our empirical findings show that CAML outperforms other meta-learners in the universal setting. Remarkably, CAML's performance in the universal setting often matches—and even exceeds—the in-domain performance of the state-of-the-art meta-learning algorithm, P>M>F (Hu et al., 2022), that is directly trained on each down-stream benchmark.

## 2 RELATED WORK

**Meta-Learning as *Causal* Sequence Modeling.** Several of the earliest meta-learning algorithms were formulated as *causal* sequence modeling problems. Hochreiter et al. (2001) leverage a LSTM (Hochreiter & Schmidhuber, 1997) to model extensions to semi-linear and quadratic functions, and two decades later, Graves et al. (2014); Santoro et al. (2016); Kaiser et al. (2017) build upon this approach by integrating a form of external memory that the LSTM can read to and write from memory to develop Neural Turing Machines. With the advent of self-attention (Vaswani et al., 2017), Mishra et al. (2017) predict the labels of query images by first composing a sequence of (image, label) pairs and then feeding it through a stack of interleaved causal self-attention and temporal convolution layers. Kirsch et al. (2022) replaces the stack of interleaved causal self-attention and temporal convolution layers with a Transformer encoder; however, their approach is also causal in the input sequence by composing a sequence of (image, label of previous image) pairs. Both Mishra et al. (2017) and Kirsch et al. (2022) are conceptually similar to our work; however, the causal property of both approaches breaks an important symmetry in meta-learning, namely invariance to permutations of the support set (Garnelo et al., 2018; Müller et al., 2021). In Section 5.2, we observe a performance gap between both approaches and CAML and hypothesize the causal approach actually forces a subtly more difficult modeling problem by imposing a causality inductive bias on a fundamentally non-causal prediction task.

**Cross-Domain Meta-Learning.** Cross-domain meta-learning refers to a challenging evaluation paradigm where the meta-training and inference-time data distributions are significantly different (Chen et al., 2019). Recent work finds that leveraging self-supervised pre-training—or foundational model feature extractors—can significantly improve cross-domain performance (Hu et al., 2022; Zhang et al., 2021). Moreover, fine-tuning with respect to the support set almost always outperforms meta-learning without fine-tuning in this setting (Guo et al., 2020; Oh et al., 2022; Phoo & Hariharan, 2020; Islam et al., 2021). While effective, fine-tuning is prohibitive to deploying visual meta-learning models in a manner similar to LLMs like ChatGPT as the latency and memory cost to fine-tune a model's parameters on each user query is untenable. Accordingly, we propose the universal setting to measure a meta-learner's ability to learn to classify *any* task seen during inference without fine-tuning.

**In-Context Learning for Dense Prediction Tasks.** Many recent works have explored in-context learning for other applications of computer vision. Bar et al. (2022) casts in-context learning as image in-painting by first concatenating demonstration images with a query image and then using a vision model to fill-in-the-blank within this concatenated image. Building on this work, Zhang et al. (2023) explores what demonstrations lead to strong in-painting performance and Wang et al. (2023) generalizes the approach by formulating other visual applications like segmentation, depth

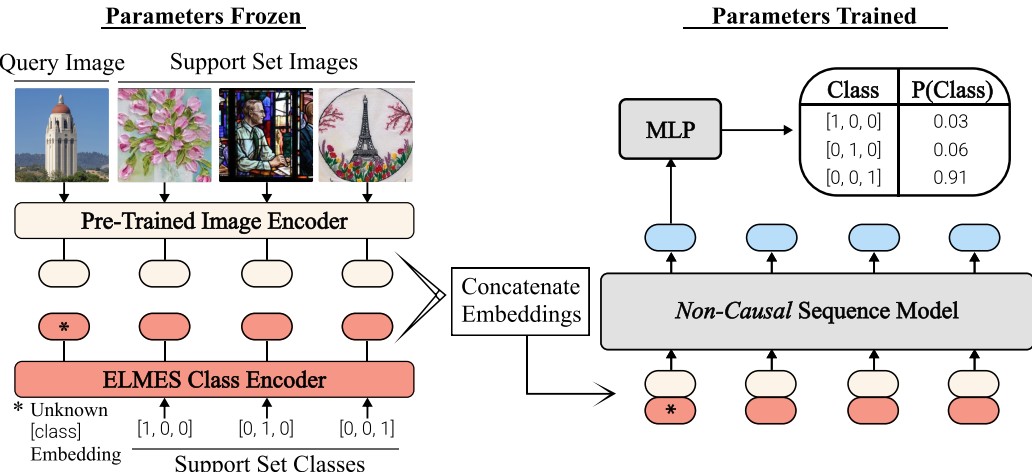

Figure 1: Overview of CAML. Query and support set images are encoded with a pre-trained feature extractor and then concatenated with their corresponding ELMES label embeddings. We feed the resulting sequence of concatenated vectors into a *non-casual* sequence model and extract the query vector from the output sequence to predict its class.

estimation, etc. as in-painting. Other approaches explore in-context learning for applications like scene understanding (Balazevic et al., 2024), medical image segmentation (Butoi et al., 2023), and more generally dense prediction tasks (Kim et al., 2023). Like these approaches, we study visual in-context learning; however, this work focuses on few-shot image classification rather than dense prediction tasks.

## 3  APPROACH

We adapt the ideas underpinning in-context learning in LLMs—namely learning to classify a query from a context of support set demonstrations in a single forward pass—to image classification. However, dissimilar from in-context learning, visual meta-learners should be non-causal: placing one example before another in the support set does not entail a causal relationship (Garnelo et al., 2018; Müller et al., 2021).

**Architecture.** An overview of CAML is shown in Figure 1. It consists of three different components: (1) a frozen pre-trained image encoder, (2) a fixed *Equal Length and Maximally Equiangular Set* (ELMES) class encoder, and (3) a *non-causal* sequence model. While pre-trained image encoders and *non-causal* sequence models are well-known, to encode label information we introduce an ELMES encoder. An ELMES encoder is a bijective mapping between the labels and a set of vectors that are equal length and maximally equiangular. Historically, labels have been encoded with one-hot vectors; however in Section 4, we prove that an ELMES encoding of mutually exclusive classes allows the sequence model to maximally identify classes within the support set.

As visualized in Figure 1, CAML first encodes query and support set images using a frozen pre-trained feature extractor. Crucially, the pre-trained image encoder's embedding space distills images into low-dimensional representations so that images with similar content and visual characteristics have similar embeddings. Classes of the support set are encoded with an ELMES class encoder; however as the class of the query is unknown, we use a special learnable "unknown token" embedding that is learned during large-scale pre-training. CAML then concatenates each image embedding with its corresponding query embedding to form an input sequence.

Progressing through Figure 1, this sequence is fed into a non-causal sequence model, i.e. a Transformer encoder, to condition the output representations on the full context of query and support set points. This enables dynamic and real-time classification; visual characteristics from query and support set images can be compared with each other to determine the specific visual features—such as content, textures, etc.—used to classify the query. From the output sequence of the non-causal sequence model, we select the element at the same position as the query in the input sequence, and pass this vector through a shallow MLP to predict the label of the query.

**Large-Scale Pre-Training.** As our focus is universal meta-learning—and CAML may encounter any new visual concept during inference—we pre-train CAML's non-causal sequence model on few-shot

image classification tasks from ImageNet-1k (Deng et al., 2009), Fungi (Schroeder & Cui, 2018), MSCOCO (Lin et al., 2014), and WikiArt (Saleh & Elgammal, 2015). We chose these datasets because they span generic object recognition (ImageNet-1k, MSCOCO), fine-grained image classification (Fungi), and unnatural image classification (WikiArt). To avoid distorting the pre-trained image encoder's embedding space, we freeze this module and only update the sequence model's parameters during pretraining. Similarly, since an ELMES minimizes the entropy of detecting classes within the support set, the label encoder is also frozen. In the context of pre-training, meta-training, and fine-tuning, CAML only requires pre-training and avoids meta-training on the train/validation splits of meta-learning benchmarks or fine-tuning on the support set during inference.

## 4 THEORETICAL ANALYSIS

In this section, we motivate our choice of the ELMES Class Encoder by considering the symmetries desirable in meta-learning algorithms. Two important symmetries are (1) invariance to the assignment of support set classes to numeric labels and (2) invariance to permutations in the ordering of the input sequence. The first invariance implies the class embeddings must be equiangular and equal norm, with an ELMES configuration minimizing the entropy of learnable model parameters detecting any given class. Later, we show an ELMES also satisfies the second symmetry. Due to space constraints, all proofs and many definitions, properties, lemmas, and theorems are allocated to Appendix A.1. We begin with a formal definition of an ELMES.

### 4.1 EQUAL LENGTH AND MAXIMALLY EQUIANGULAR SET OF VECTORS

**Definition 1.** *An **Equal Length and Maximally Equiangular Set (ELMES)** is a set of non-zero vectors $\{\phi_j\}_{j=1}^d$, $\phi_j \in \mathbb{R}^{d+k}$ for some $k \geq 0$ and $d > 1$, such that $\forall j \neq j'$, $\|\phi_j\| = \|\phi_{j'}\|$ and $\langle \phi_j , \phi_{j'} \rangle = \frac{-1}{d-1}$. Simply, all vectors in this set are equal length and maximally equiangular.*

An *Equal Angle and Maximally Equiangular Set* (ELMES) of vectors has connections to both Equiangular Tight Frames in representation theory (Welch, 1974; Fickus et al., 2018) as well as the Simplex Equiangular Tight Frames highlighted in recent neural collapse works exploring softmax-layer geometry at the terminal phase of training (Papyan et al., 2020; Yang et al., 2022). We offer additional discussion comparing these structures in Appendix A.1 as well as provide an intuitive view of an ELMES as a regular $d$-simplex immersed in $\mathbb{R}^{d+k}$.

### 4.2 LABEL SYMMETRY

Symmetry in the assignment of support classes to numeric labels is an important property of meta-learning algorithms. For example, if we have the support set classes {tower, bear, tree}, the mapping of {bear -> 1, tower -> 2, tree -> 3} should produce the same prediction for a query point as a different mapping {bear -> 2, tower -> 3, tree -> 1}. To explore this symmetry, we examine how class embeddings may be used by the model.

From our formulation in Section 3, we represent a demonstration vector as a concatenation of an image embedding $\rho$ and a label embedding $\phi$: $[\rho \mid \phi]$. This vector is directly fed into the self-attention mechanism, where we matrix multiply with key, query, and value self-attention heads. Taking only one of these matrices for simplicity with head-dimension $k$:

$$[\rho \mid \phi] \begin{bmatrix} \Gamma_1 & ... & \Gamma_k \\ \psi_1 & ... & \psi_k \end{bmatrix} = [\langle \rho , \Gamma_1 \rangle \quad ... \quad \langle \rho , \Gamma_k \rangle] + [\langle \phi , \psi_1 \rangle \quad ... \quad \langle \phi , \psi_k \rangle] \quad (1)$$

The output of this transformation will be the sum of two vectors: one composed of the inner products between the image embedding $\rho$ and the learnable $\{\Gamma_i\}_{i=1}^k$ and the other composed of the class embedding $\phi$ and the learnable $\{\psi_i\}_{i=1}^k$. Note that Equation (1) implies that CAML is not invariant to the assignment of labels to support set classes due to the addition between $\langle \rho , \Gamma_i \rangle$ and $\langle \phi , \psi_i \rangle$; however, we can constrain the geometry of the class embeddings $\{\phi\}_{j=1}^d$ to in principle respect label symmetry. Specifically for $i \neq j \neq k$, $\langle \phi_i , \phi_j \rangle = \langle \phi_i , \phi_k \rangle$ and $\|\phi_i\| = \|\phi_j\|$.

Similar to a convolutional filter learning to match a pattern within an image, our analysis assumes the learnable $[\psi_1 \quad ... \quad \psi_k]$ will converge to vectors that maximize the inner product with a single

class embedding subject to certain constraints. Under this assumption, we ask what geometry of the $d$-class embeddings $\{\phi\}_{j=1}^{d}$ allows a learnable $\psi_i$ vector to most unambiguously detect a single class embedding. To answer this question, we define a probability mass function for each $\psi_i$ over the set of $d-$classes so that maximizing the probability of the $j^{th}$ class aligns with maximizing $\langle \phi_j , \psi_i \rangle$ and equally minimizing $\langle \phi_k , \psi_i \rangle$ for $k \neq j$.

**Definition 2.** *Let $X$ be a discrete Random Variable taking on values in $\{1, 2, ..., d\}$. For learnable vector $\psi_i$, define probability mass function $p_{\psi_i}(X = j)$ as the softmax over $[\langle \phi_1 , \psi_i \rangle \quad ... \quad \langle \phi_d , \psi_i \rangle]$ so that:*

$$p_{\psi_i}(X = j) = \frac{e^{\|\psi_i\|\|\phi_j\|\cos(\theta_{i,j})}}{\sum_{k=1}^{d} e^{\|\psi_i\|\|\phi_j\|\cos(\theta_{i,k})}}$$

*where $\theta_{i,j}$ is the angle between $\phi_j$ and $\psi_i$.*

We say $\psi_i$ learns to detect class $j$ when $p_{\psi_i}(X = j) > p_{\psi_i}(X = k)$ for $1 \leq k \leq d$ with $k \neq j$. By symmetry in the assignment of class embeddings to support classes, we can assume that the number of $\psi_i$ learned to detect class $i$ is similar to the number of $\psi_j$ learned to detect class $j$ for all pairs $(i, j)$. We also leverage symmetry in the assignment of labels to support set classes to make the following assumptions. A justification for each assumption is located in Appendix A.1.

**Assumption 1.** *Suppose $\{\psi_i\}_{i=1}^{k}$ are learnable class detectors of unit norm with at least one $\psi_i$ detecting each class $1 \leq i \leq d$. The probability $p_{\psi_j}(X = j) = p_{\psi_i}(X = i)$ for $1 \leq i, j \leq d$.*

**Assumption 2.** *Define $p_{\psi_i}(X = i)\backslash\{\phi_l\}_{l=(m+1)}^{d}$ as the probability of $\psi_i$ detecting $\phi_i$ from the set of vectors $\{\phi_j\}_{j=1}^{m}$, $m < d$. Then the probability $p_{\psi_j}(X = j)\backslash\{\phi_l\}_{l=(m+1)}^{d} = p_{\psi_i}(X = i)\backslash\{\phi_l\}_{l=(m+1)}^{d}$ for $1 \leq i, j \leq m$ and $m \geq 2$.*

**Assumption 3.** *When $\psi_i = \frac{\phi_i}{\|\phi_i\|}$, $p_{\psi_i}(X = i)$ is maximized.*

When Assumption 1, Assumption 2, and Assumption 3 hold, the set of class embeddings that maximize the probability of a learnable $\psi_i$ detecting class $i$ is necessarily an ELMES.

**Theorem 1.** *The set of class embeddings $\{\phi_j\}_{j=1}^{d} \; \forall j, \; 1 \leq j \leq d$ that maximizes $p_{\psi_j}(X = j)$ is necessarily an ELMES.*

Alternatively when viewed through the lens of information theory, we can interpret an ELMES as the class embedding that minimizes the entropy of $\psi_i$ detecting class $i$. Informally, ELMES causes $\psi_i$ to have the least uncertainty when detecting class $i$.

**Proposition 1.** *Let $H_{\psi_i}(X)$ be the entropy of $p_{\psi_i}(X)$. An ELMES minimizes $H_{\psi_i}(X)$.*

### 4.3 PERMUTATION INVARIANCE.

In addition to label symmetry, it is also desirable for the output prediction of CAML to not depend on the order of demonstrations in the sequence. For example, if we have the support set classes {tower, bear, tree}, the sequence {(bear -> 1), (tower -> 2), (tree -> 3)} should produce the same output as the permuted sequence {(tree -> 3), (bear -> 1), (tower -> 2)}. Building on the prior work of Kossen et al. (2021); Fifty et al. (2023), it suffices to show to show that the ELMES label encoder is equivariant to permutations in the input sequence to show that CAML is invariant to permutations.

**Proposition 2.** *Consider an $n$-sequence of one-hot labels stacked into a matrix $\mathcal{S} \in \mathbb{R}^{n \times w}$, and an ELMES label encoder denoted by $W \in \mathbb{R}^{w \times d}$ with $w$ denoting "way" and $d$ the dimension of the label embedding. The label embedding $\mathcal{S}W$ is equivariant to permutations.*

## 5 EXPERIMENTS

To quantify universal image classification performance, we evaluate a diverse set of 11 meta-learning benchmarks divided across 4 different categories:

1. Generic Object Recognition: mini-ImageNet (Vinyals et al., 2016), tiered-ImageNet (Ren et al., 2018), CIFAR-fs (Bertinetto et al., 2018), and Pascal VOC (Everingham et al.)

Table 1: **MiniImageNet & CIFAR-fs** mean accuracy and standard error across 10,000 test epochs. †
indicates the pre-trained image encoder backbone was frozen during training.

| Method (Backbone) | CIFAR-fs | | MiniImageNet | |
|---|---|---|---|---|
| | 5w-1s | 5w-5s | 5w-1s | 5w-5s |
| **In-Domain [Meta-Training]** | | | | |
| P>M>F Hu et al. (2022) | 84.3 | 92.2 | 95.3 | 98.4 |
| **Universal Meta-Learning;** | | | | |
| **No Meta-Training or Finetuning** | | | | |
| ProtoNet (Snell et al., 2017) | 62.9±.2 | 79.7±.2 | 92.1±.1 | 97.1±.0 |
| ProtoNet† | 57.7±.2 | 81.0±.2 | 85.3±.2 | 96.0±.1 |
| MetaOpt (Lee et al., 2019) | 53.1±.3 | 73.1±.2 | 78.5±.2 | 91.6±.1 |
| MetaOpt† | 61.7±.2 | 83.1±.1 | 86.9±.2 | 96.5±.1 |
| MetaQDA (Zhang et al., 2021) | 60.4±.2 | 83.2±.1 | 88.2±.2 | 97.4±.0 |
| GPICL (Kirsch et al., 2022) | 41.5±.4 | 78.3±.2 | 95.6±.1 | 98.2±.1 |
| SNAIL (Mishra et al., 2017) | 62.1±.3 | 71.1±.3 | 93.6±.1 | 98.1±.0 |
| CAML | **70.8**±.2 | **85.5**±.1 | **96.2**±.1 | **98.6**±.0 |

Table 2: **Pascal & Paintings** mean accuracy and standard error across 10,000 test epochs. † indicates
the the pre-trained image encoder backbone was frozen during training.

| Method (Backbone) | Pascal + Paintings | | Paintings | | Pascal | |
|---|---|---|---|---|---|---|
| | 5w-1s | 5w-5s | 5w-1s | 5w-5s | 5w-1s | 5w-5s |
| **In-Domain [Meta-Training]** | | | | | | |
| P>M>F | 60.7 | 74.4 | 53.2 | 65.8 | 72.2 | 84.4 |
| **Universal Meta-Learning** | | | | | | |
| ProtoNet | 49.6±.2 | 63.5±.1 | 38.3±.2 | 48.2±.1 | 77.9±.2 | 87.3±.2 |
| ProtoNet† | 52.2±.2 | 70.6±.1 | 48.3±.2 | 64.1±.1 | 72.2±.2 | 84.3±.2 |
| MetaOpt | 38.2±.2 | 58.2±.1 | 31.6±.2 | 48.0±.1 | 63.7±.2 | 81.7±.2 |
| MetaOpt† | 53.2±.2 | 74.8±.1 | 49.3±.2 | 65.9±.1 | 72.8±.2 | 84.4±.2 |
| MetaQDA | 53.8±.2 | 74.1±.1 | 49.4±.2 | **66.6**±.1 | 73.5±.2 | 85.2±.2 |
| GPICL | 62.6±.2 | 74.6±.1 | 51.6±.2 | 61.0±.1 | 81.7±.2 | 88.2±.2 |
| SNAIL | 62.5±.2 | 77.6±.1 | **51.9**±.2 | 65.8±.1 | 79.7±.2 | 88.0±.2 |
| CAML | **63.8**±.2 | **78.3**±.1 | 51.1±.2 | 65.2±.1 | **82.6**±.2 | **89.7**±.1 |

2. Fine-Grained Image Classification: CUB (Wah et al., 2011), Aircraft (Maji et al., 2013), meta-
   iNat (Wertheimer & Hariharan, 2019), and tiered meta-iNat (Wertheimer & Hariharan, 2019)

3. Unnatural Image Classification: ChestX (Guo et al., 2020) and Paintings (Crowley & Zisserman,
   2015)

4. Inter-Domain Image Classification: Pascal+Paintings (Everingham et al.; Crowley & Zisserman,
   2015).

Generic object recognition, fine-grained image classification, and unnatural image classification
are standard benchmarking tasks in meta-learning literature (Chen et al., 2020; Hu et al., 2022;
Wertheimer et al., 2020; Guo et al., 2020). Beyond this, we compose a challenging new *inter-domain*
category by combining Pascal VOC with Paintings so that each class is composed of both natural
images and paintings. This allows us to evaluate the ability of meta-learning algorithms to generalize
across domains within the same class. For example, the support image for the class "tower" may be
Van Gogh's *The Starry Night*, while the query may be a picture of the Eiffel Tower. Humans have
the ability to generalize visual concepts between such domains; however, meta-learning algorithms
struggle with this formulation (Jankowski & Grąbczewski, 2011).

## 5.1 BASELINES

We evaluate the performance of CAML, Prototypical Networks (ProtoNet) (Snell et al., 2017),
MetaOpt (Lee et al., 2019), MetaQDA (Zhang et al., 2021), SNAIL (Mishra et al., 2017), and
GPICL (Kirsch et al., 2022) in a universal meta-learning setting by pre-training them with a ViT-
base (Dosovitskiy et al., 2020) feature extractor initialized with weights from CLIP (Radford et al.,

Table 3: **meta-iNat & tiered meta-iNat & ChestX** mean accuracy and standard error across 10,000 test epochs. † indicates the the pre-trained image encoder backbone was frozen during training.

| Method (Backbone) | meta-iNat | | tiered meta-iNat | | ChestX | |
|---|---|---|---|---|---|---|
| | 5w-1s | 5w-5s | 5w-1s | 5w-5s | 5w-1s | 5w-5s |
| **In-Domain [Meta-Training]** | | | | | | |
| P>M>F | 91.2 | 96.1 | 74.8 | 89.9 | 27.0 | 32.1 |
| **Universal Meta-Learning** | | | | | | |
| ProtoNet | 78.4±.2 | 89.4±.1 | 66.3±.2 | 82.2±.2 | 22.4±.1 | 25.3±.1 |
| ProtoNet† | 84.5±.2 | 94.8±.1 | 73.8±.2 | 89.5±.1 | 22.7±.1 | 25.8±.1 |
| MetaOpt | 53.0±.2 | 77.7±.2 | 37.3±.2 | 63.0±.2 | 20.8±.1 | 23.0±.1 |
| MetaOpt† | 85.5±.2 | 95.5±.1 | 75.1±.2 | 91.9±.1 | **23.0±.1** | **27.4±.1** |
| MetaQDA | 86.3±.2 | 95.9±.1 | 76.0±.2 | **92.4±.1** | 22.6±.1 | 27.0±.1 |
| GPICL | 90.0±.2 | 95.1±.1 | 60.8±.5 | 87.6±.2 | 20.1±.1 | 20.9±.1 |
| SNAIL | 89.1±.2 | 94.8±.1 | 77.3±.2 | 86.5±.2 | 20.2±.0 | 20.0±.0 |
| CAML | **91.2±.2** | **96.3±.1** | **81.9±.2** | 91.6±.1 | 21.5±.1 | 22.2±.1 |

Table 4: **CUB & tiered-ImageNet & Aircraft** mean accuracy and standard error across 10,000 test epochs. † indicates the the pre-trained image encoder backbone was frozen during training.

| Method (Backbone) | CUB | | tiered-ImageNet | | Aircraft | |
|---|---|---|---|---|---|---|
| | 5w-1s | 5w-5s | 5w-1s | 5w-5s | 5w-1s | 5w-5s |
| **In-Domain [Meta-Training]** | | | | | | |
| P>M>F | 92.3 | 97.0 | 93.5 | 97.3 | 79.8 | 89.3 |
| **Universal Meta-Learning** | | | | | | |
| ProtoNet | 59.4±.2 | 77.3±.2 | 93.5±.1 | 97.4±.1 | 37.9±.2 | 52.5±.2 |
| ProtoNet† | 87.0±.2 | **97.1±.1** | 87.3±.2 | 96.1±.1 | 62.4±.3 | 82.0±.2 |
| MetaOpt | 71.5±.2 | 41.2±.2 | 76.6±.2 | 89.6±.1 | 41.6±.2 | 26.7±.1 |
| MetaOpt † | 87.9±.2 | **97.2±.1** | 88.2±.2 | 96.5±.1 | **64.8±.2** | **82.6±.2** |
| MetaQDA | 88.3±.2 | **97.4±.1** | 89.4±.2 | 97.0±.1 | 63.6±.3 | **83.0±.2** |
| GPICL | 75.1±.5 | 94.5±.1 | 94.6±.1 | 97.2±.1 | 19.8±.2 | 61.8±.3 |
| SNAIL | 87.5±.2 | 92.8±.2 | 93.1±.1 | 97.3±.1 | 48.9 ± .3 | 35.8±.3 |
| CAML | **91.8±.2** | **97.1±.1** | **95.4±.1** | **98.1±.1** | 63.3±.3 | 79.1±.2 |

2021). Pre-training runs over few-shot classification tasks from ImageNet-1k, Fungi, MSCOCO, and WikiArt, and during evaluation on the set of 11 meta-learning benchmarks, models are not meta-trained or fine-tuned. We compare with ProtoNet, MetaOpt, and MetaQDA as they achieve state-of-the-art results when paired with a pre-trained feature extractor (Hu et al., 2022). As sequence modeling underpins CAML, we also compare with SNAIL and GPICL to evaluate the performance of past formulations of *causal* sequence-based meta-learning algorithms in the universal setting.

To assess the gap between universal and in-domain meta-learning performance, we benchmark the current state-of-the-art meta-learning algorithm P>M>F (Hu et al., 2022). Similar to the universal setting, P>M>F uses a ViT-base feature extractor initialized with weights from DINO (Caron et al., 2021); however, it meta-trains on the training set of each benchmark before evaluating on that benchmark's test set.

When pre-training all models in the universal setting, we set the learning rate to a fixed $1 \times 10^{-5}$ and do not perform any hyperparameter tuning in order to match the practices used by P>M>F. We use early stopping with a window size of 10 epochs during pre-training and the code release of Hu et al. (2022) to benchmark P>M>F with the training settings and hyperparameters described in their work.

## 5.2 RESULTS

Our findings are summarized in Table 1, Table 2, Table 3, and Table 4 and indicate that CAML sets a new state-of-the-art for universal meta-learning by significantly outperforming other baselines on 14 of 22 evaluation settings. For 5 of the other 8 evaluation settings, CAML matches—or nearly matches—the best performing baseline. Remarkably, CAML also performs competitively with

P>M>F on 8 out of 11 meta-learning benchmarks, even though P>M>F meta-trains on the training set of each benchmark.

This result suggests that the amount of new visual information learned during inference through visual in-context learning can be comparable to the amount learned when directly meta-training on in-domain data. This capacity may unlock new applications in the visual space, just as the emergence of in-context learning in LLMs has enabled many new applications in natural language.

**Benchmarks Where CAML Underperforms.** The 3 datasets where P>M>F outperforms CAML are CIFAR-fs, Aircraft, and ChestX. CIFAR-fs is a generic object recognition benchmark containing CIFAR images downsampled to 32x32 resolution. As CAML and CLIP pre-train on 224x224 resolution images, downsampling by a factor of 49 likely induces a distribution shift that was not learned by CAML during large-scale pre-training. In the cases of Aircraft and ChestX, we postulate that the CLIP embedding space—structured so images with similar captions have similar embeddings–struggles to effectively differentiate between the fine-grained, specialized classes in these tasks. For example, while a Boeing 737 and Airbus A380 have different labels in the Aircraft dataset, the scraped CLIP captions for those images may not reach that level of granularity. This corroborates the findings from Radford et al. (2021), which found that in a zero-shot setting, CLIP underperforms in specialized or complex tasks.

Our ablation study into non-CLIP pre-trained feature extractors in Tables 5 to 8 of Appendix C shows CAML's performance on Aircraft can drastically improve. Specifically, 5w-1s performance increases from 63.3 to 81.8 and 5w-5s performance increases from 79.1 to 92.1 when a ViT-Huge pre-trained on Laion-2b (Schuhmann et al., 2022) initializes the weights of the image encoder rather than CLIP.

**Fine-tuning CLIP Backbone.** Our findings in Tables 1 to 4 indicate that updating the CLIP image encoder during pre-training hurts the performance of ProtoNet and MetaOpt. We observe that these methods tend to overfit during pre-training, and our empirical results show a similar pattern: pre-training with these methods often helps the performance of benchmarks similar to ImageNet (i.e. Pascal, MiniImageNet, tiered-ImageNet), but it significantly hurts the performance of out-of-domain tasks (i.e. Aircraft, CUB, Paintings) as shown in Tables 1 to 4. We believe that further training the CLIP backbone distorts the structure of its embedding space, leading to catastrophic forgetting on out-of-domain tasks. Conversely, CAML, MetaQDA, SNAIL, and GPICL—all of which freeze the parameters of the CLIP feature extractor—benefit greatly from large-scale episodic pre-training on ImageNet-1k, Fungi, MSCOCO, and WikiArt.

## 6 ANALYSIS

To better understand how CAML learns during inference, we analyze its ability to dynamically update its representations. Due to casting meta-learning as *non-causal* sequence modeling, CAML considers the full context of query and support set to predict the label of the query. Specifically, the query dynamically influences the representation of support set points, and the support set points dynamically influence the representation of the query as this sequence is passed through the layers of a non-causal sequence model. This property enables universal meta-learning by allowing the model to update the support and query representations based on the context of the task, not only the contents of the images, within the parameter space of the sequence model.

An example where the query dynamically influences the support set is visualized in Figure 2. Given only the 5 support examples, the prediction task is ambiguous. However, the nature of the query determines the prediction task. The query image of a tower in Figure 2a reduces the task to generic object recognition: classify the query based on the object portrayed in the image. On the other hand, and as visualized in Figure 2b, the query image of embroidery reduces the prediction task to texture identification: classify the query based on artistic medium.

To analyze how dynamic representations affect CAML, we examine the representations of the support set and query vectors at the input to and output of the non-causal sequence model. For both examples visualized in Figure 2a and Figure 2b, the non-causal sequence model learns to separate support set vectors by class identity and group the query representation with the correct support set example.

We find the frozen CLIP image embeddings are actually antagonistic for the classification-by-texture task visualized in Figure 2b: the query image embedding is closest to the support set example for

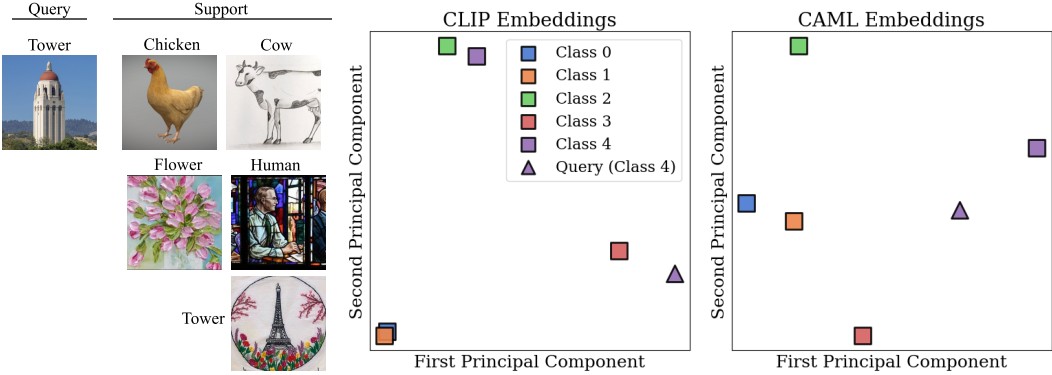

(a) Left: An example task—classify images by the objects depicted. Center: image embeddings output from the Image Encoder (CLIP) in CAML . Right: joint image-label representations output by the non-causal sequence model in CAML for the same task.

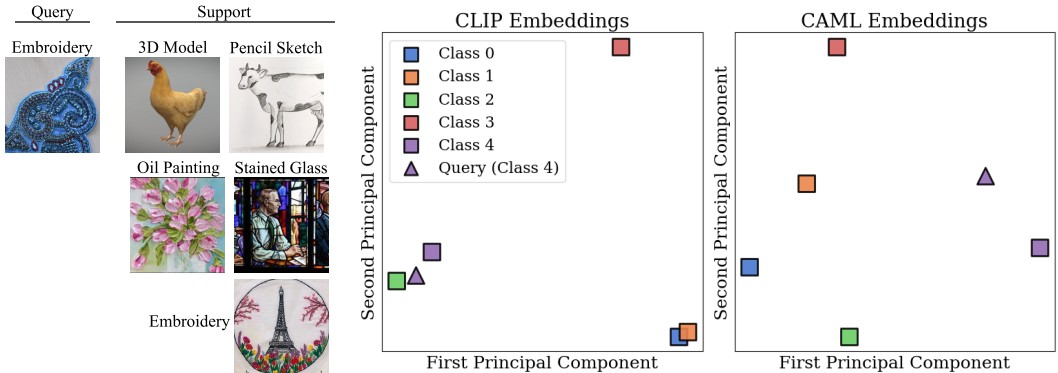

(b) Left: An example task—classify images by the artistic medium used. Center: CLIP image embeddings output from the Image Encoder (CLIP) in CAML . Right: joint image-label representations output by the non-causal sequence model in CAML for the same task.

Figure 2: Two sample tasks over the same support images but utilizing different criteria to define classes. The nature of the query image informs the task being presented, e.g. classification by object (top) vs. classification by texture (bottom). For both tasks, the output of the non-causal sequence model provides better separation among class representations than CLIP embeddings and groups the query representation with the proper task, even when projected into 2D space by PCA.

the second class, "oil painting". Unsurprisingly, the baseline methods that rely on frozen CLIP embeddings—specifically MetaQDA, ProtoNet[†], and MetaOpt[†]—group the query with "oil painting" and therefore misclassify this example. On the other hand, as CAML considers the full context of the query and support set, it develops representations of the query in the context of the support set—and the support set in the context of the query—to group the query with the "embroidery" support set image as they share the same texture, thereby correctly classifying this example.

# 7   CONCLUSION

In this work, we develop *universal meta-learning* to approximate the performance of visual meta-learners deployed to a ChatGPT-like application and present CAML: a meta-learning algorithm that emulates in-context learning in LLMs by learning new visual concepts during inference without fine-tuning. Our empirical findings show that CAML—without meta-training or fine-tuning—exceeds or matches the performance of the current state-of-the-art meta-learning algorithm on 8 out of 11 benchmarks. This result indicates visual meta-learning models are ready for deployment in a manner similar to LLMs, and we hope this work recalibrates our sense of limitations for the universal meta-learning paradigm.

Nevertheless, there are areas where CAML struggles. Specifically, the performance of CAML on highly out-of-distribution images—e.g. chest x-rays—and varying image resolutions—e.g. rescaled CIFAR images—lags behind that of the best *in-domain* approaches. Developing methods for the *universal* setting that are robust to these cases is a promising direction for future work.

ACKNOWLEDGMENTS

We thank Mayee Chen, Dan Fu, Jerry Liu, and Benjamin Spector for their invaluable feedback and help during revisions of this work. We also thank Chelsea Finn for helping us improve the related work, Victor Butoi for constructive dialogue over Twitter, and the Hazy Group at Stanford as a whole their support throughout the research process. We gratefully acknowledge the support of NIH under No. U54EB020405 (Mobilize), DARPA under Nos. N660011924033 (MCS), NSF under Nos. OAC-1835598 (CINES), CCF-1918940 (Expeditions), DMS-2327709 (IHBEM), Nos. CCF2247015 (Hardware-Aware), CCF1763315 (Beyond Sparsity), CCF1563078 (Volume to Velocity), and 1937301 (RTML); US DEVCOM ARL under Nos. W911NF-23-2-0184 (Long-context) and W911NF-21-2-0251 (Interactive Human-AI Teaming); ONR under Nos. N000142312633 (Deep Signal Processing); Stanford HAI under No. 247183; NXP, Xilinx, LETI-CEA, Intel, IBM, Microsoft, NEC, Toshiba, TSMC, ARM, Hitachi, BASF, Accenture, Ericsson, Qualcomm, Analog Devices, Google Cloud, Salesforce, Total, Wu Tsai Neurosciences Institute, Chan Zuckerberg Initiative, Amazon, Genentech, GSK, Juniper Networks, KDDI, UCB, the HAI-GCP Cloud Credits for Research program, the Stanford Data Applications Initiative, and the Stanford Data Science Initiative (SDSI). The U.S. Government is authorized to reproduce and distribute reprints for Governmental purposes notwithstanding any copyright notation thereon. Any opinions, findings, and conclusions or recommendations expressed in this material are those of the authors and do not necessarily reflect the views, policies, or endorsements, either expressed or implied, of NIH, ONR, or the U.S. Government.

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

# A APPENDIX

## A.1 SUPPLEMENTARY THEORETICAL ANALYSIS

We offer additional insight into the theoretical analysis presented in Section 4 and provide the omitted remarks, properties, lemmas, and proofs.

### A.1.1 EQUIANGULAR TIGHT FRAMES

Papyan et al. (2020) coin the term Simplex Equianguar Tight Frame to describe a set of vectors $\{\phi_j\}_{j=1}^d$ such that the minimum angle between any two pairs of vectors is maximized and all vectors have equal norm. Formally,

**Definition 3.** *Let $\mathbb{R}^d$ be a $d-$dimensional inner product space over $\mathbb{R}$ with the Euclidean inner product. A **Simplex ETF** is a set of d vectors $\{\phi_j\}_{j=1}^d$, $\phi_j \in \mathbb{R}^d$, specified by the columns of*

$$\sqrt{\frac{d}{d-1}}(I_d - \frac{1}{d}\mathbb{1}\mathbb{1}^T)$$

where $I_d \in \mathbb{R}^{d \times d}$ is the identity matrix and $\mathbb{1} \in \mathbb{R}^{d \times 1}$ is the ones vector. Somewhat contradictory, a Simplex Equiangular Tight Frame is not an Equiangular Tight Frame (Welch, 1974) as this set of vectors does not form a tight frame in $\mathbb{R}^d$.

**Definition 4.** *Let $\mathbb{R}$ be a $d-$dimensional space over $\mathbb{R}$ with the Euclidean inner product. An **Equiangular Tight Frame (ETF)** is a set of non-zero, equal norm vectors $\{\phi_j\}_{j=1}^n$, $n \geq d$, that achieves the Welch lower bound:*

$$\max_{j \neq j'} \frac{|\langle \phi_j , \phi_{j'} \rangle|}{\|\phi_j\|\|\phi_{j'}\|} = \sqrt{\frac{n-d}{d(n-1)}}$$

It is well-known that a set of non-zero equal-norm vectors satisfies the Welch lower bound if and only if that set of vectors is equiangular and also a tight frame for $\mathbb{R}^d$ (Fickus et al., 2018).

**Definition 5.** *A set of non-zero, equal norm vectors $\{\phi_j\}_{j=1}^n$ is **equiangular** if $\forall j \neq j'$, $|\langle \phi_j , \phi_{j'} \rangle| = c$ for some $c \in \mathbb{R}$, $c > 0$.*

**Definition 6.** *$\{\phi_j\}_{j=1}^n$ is a **tight frame** for $\mathbb{R}^d$ if, $\forall v \in \mathbb{R}^d$, $\exists A > 0$ such that $A\|v\|^2 = \sum_{j=1}^n |\langle \phi_j , v \rangle|^2$.*

**Remark 1.** *A Simplex Equiangular Tight Frame is not a tight frame.*

*Proof.* Observe that for any finite $d$, for $\{\phi_j\}_{j=1}^d$ equal to the columns of $\sqrt{\frac{d}{d-1}}(I_d - \frac{1}{d}\mathbb{1}\mathbb{1}^T)$, it is the case that $\sum_{j=1}^{d-1} \phi_j = -1 * \phi_d$. So $\{\phi_j\}_{j=1}^n$ do not span $\mathbb{R}^d$, and therefore, cannot be a tight frame. $\square$

Similarly, a Simplex ETF is not a $d-$simplex.

**Remark 2.** *A Simplex Equiangular Tight Frame is not a simplex.*

*Proof.* A simplex in $\mathbb{R}^n$ requires $n + 1$ points. $\square$

To align terminology with properties, we generalize a Simplex ETF to an ELMES in Definition 1: a set of $d$ vectors in a $(d + k)$-dimensional ambient space with $k \geq 0$. Observe that a regular simplex is a special type of ETF in which the number of vectors in the set is one more than the dimension of the space that they span (Fickus et al., 2018). Building on this observation, an intuitive view of ELMES is a regular $d-$simplex immersed in $\mathbb{R}^{d+k}$.

**Remark 3.** *Consider a centered d-dimensional regular simplex with vertices $\{\phi_j\}_{j=1}^{d+1}$, $\phi_j \in \mathbb{R}^{d+1}$. Let $\iota_{can}$ be the canonical inclusion map: $\mathbb{R}^d \to \mathbb{R}^{d+1}$, $\iota_{can}(x_1, x_2, ..., x_d) = (x_1, x_2, ..., x_d, 0_{d+1})$, then $\{\iota_{can}(\phi_j)\}_{j=1}^{d+1}$ is an ELMES.*

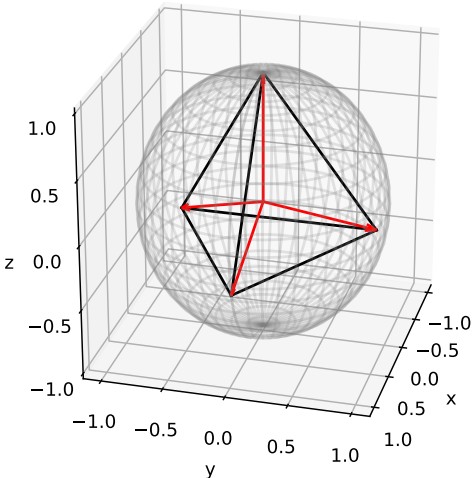

Figure 3: A visualization of a $d = 4$ ELMES in $\mathbb{R}^3$. Observe the endpoints of the vectors of an ELMES lie on the vertices of a centered regular tetrahedron.

*Proof.* The two criteria of an ELMES are maximally equiangular and equal length. As all vertices of a centered regular $d-$simplex are equal length from the origin, $\{\phi_j\}_{j=1}^{d+1}$ are equal length and therefore $\{\imath_{can}(\phi_j)\}_{j=1}^{d+1}$ must also have equal length.

Similarly, from Lemma 10 of Papyan et al. (2020), we know the cosine of the angle between any two vectors in a $(d+1)-$dimensional ELMES is $\frac{-1}{d}$. It is known that for a $d-$dimensional regular simplex in $\mathbb{R}^d$ centered at the origin, the angle subtended by any two verticies through the origin is $\cos(\theta) = \frac{-1}{d}$. Immersing $\{\phi_j\}_{j=1}^{d+1}$, $\phi_j \in \mathbb{R}^d$, into $\mathbb{R}^{d+1}$ via the canonical inclusion operator $\imath_{can}$ does not change the pairwise angle between vectors in this set: $\langle \phi_j , \phi_{j'} \rangle = \langle \imath_{can}(\phi_j) , \imath_{can}(\phi_{j'}) \rangle$. As $\{\imath_{can}(\phi_j)\}_{j=1}^{d+1}$ are equal length and maximally equiangular, it forms an ELMES. □

We now show that an ELMES immersed in a higher dimension remains an ELMES. Taken with Remark 3, we can view a high-dimensional ELMES in $\mathbb{R}^d$ composed of $n + 1$ vectors $\{\phi_j\}_{j=1}^{n+1}$, $d >> n + 1$, as simply a $n-$simplex immersed in $\mathbb{R}^d$ via the canonical inclusion operator.

**Lemma 1.** *Let $\imath_{can} : \mathbb{R}^d \to \mathbb{R}^{d+k}$. If $\{\phi_j\}_{j=1}^n$ is an ELMES , then $\{\imath_{can}(\phi_j)\}_{j=1}^d$ is an ELMES.*

*Proof.* This reduces to proving that the maximum angle between a set of $d$ equiangular points in $\mathbb{R}^d$ is the maximum angle between a set of $d$ equiangular points in $\mathbb{R}^{d+k}$. Let $\{\phi_j\}_{j=1}^d$ be an ELMES such that $\phi_j \in \mathbb{R}^d$ and $\{\psi_j\}_{j=1}^d$ be an ELMES such that $\psi_j \in \mathbb{R}^{d+k}$. Then $\{\psi_j\}_{j=1}^d$ lie in a $d-$dimensional subspace of $\mathbb{R}^{d+k}$: $\exists \gamma_1, ..., \gamma_d$ and basis vectors $e_1, ..., e_d$ such that $\forall \psi_j \in \{\psi_j\}_{j=1}^d$, $\psi_j = \sum_{i=1}^d \gamma_i e_i$. Therefore, $\forall j \neq j'$, $\langle \psi_j , \psi_{j'} \rangle \leq \langle \phi_j , \phi_{j'} \rangle$ as $\{\phi_j\}_{j=1}^d$ are an ELMES for $\mathbb{R}^d$. □

### A.1.2 ELMES ROTATIONAL SYMMETRY

There are infinitely many ELMES by rotating one such set of vectors about the origin.

**Remark 4.** *Let $\{\phi_j\}_{j=1}^d$ be an ELMES in $\mathbb{R}^{d+k}$ for some $k \geq 0$. Let $o : \mathbb{R}^{d+k} \to \mathbb{R}^{d+k}$ be an operator from the special orthogonal group $SO(d + k)$. Then $\{o(\phi_j)\}_{j=1}^d$ is also an ELMES .*

*Proof.* Length is preserved as operations in $SO(d + k)$ have determinant 1 and angles are similarly preserved as operations in $SO(d + k)$ are unitary (i.e. preserving inner product). □

### A.1.3 A SET OF ORTHONORMAL BASIS VECTORS IS NOT AN ELMES

A final remark relates to the common misconception that a set of orthonormal basis vectors $\{\psi_j\}_{j=1}^d$ is an ELMES. While $\{\psi_j\}_{j=1}^d$ is an ETF in $\mathbb{R}^d$ since this set realizes the Welch lower-bound in Definition 4, these vectors are not maximally equiangular: $\langle \psi_j , \psi_{j'} \rangle = 0 > \frac{-1}{d-1}$.

### A.2 ELMES MAXIMIZES $p_{\psi_j}(X = j)$

*Justification of Assumption 1.* This property is implied by symmetry in the assignment of class embeddings to support classes. As the assignment is arbitrary, all learnable $\psi_i$ class detectors should have equal probability of detecting their respective class. For simplicity of notation, we say $\psi_i$ learns to detect class embedding $\phi_i$ rather another class embedding $\phi_k$, $k \neq i$. □

*Justification of Assumption 2.* Informally, this property states that, for any $m$-subset of classes $\{\phi_j\}_{j=1}^m$, the probability of $\psi_j$ detecting class $j$ is equal to the probability of $\psi_i$ detecting class $i$. This is again implied by symmetry in the assignment of class embeddings to support classes as meta-learning algorithms may predict among a subset of $m$ classes in the support set rather than the maximum number of classes $d$. □

*Justification of Assumption 3.* Recall in $\mathbb{R}^d$, $\langle \psi , \phi \rangle = \|\psi\|\|\phi\| \cos(\theta)$ where $\theta$ is the angle between $\psi_i$ and $\phi_i$. Then this assumption constrains our set $\{\phi_j\}_{j=1}^d$ so that relative norm of $\phi_i$ with respect to $\phi_j$ is lower bounded by $\cos(\theta_{i,j})$: $\frac{\|\phi_i\|}{\|\phi_j\|} > \cos(\theta_{i,j})$.

Informally, the $\{\phi_j\}_{j=1}^d$ are sufficiently spread out in the ambient space so that the learnable $\psi_i$ that maximizes $p_{\psi_i}(X = i)$ is $\phi_i$ itself: $\psi_i = \frac{\phi_i}{\|\phi_i\|}$. This constraint helps us avoid degenerative cases like $\{\phi_j\}_{j=1}^d$ all equal. For example, $\phi_j = \alpha\phi_i$, $i \neq j$ with $\alpha > 0$ is one such degenerative case where one class embedding vector is stacked on a different class embedding, but with higher norm. □

*Proof of Theorem 1.* Taken with Assumption 1, Assumption 2, and Assumption 3, it suffices to show Theorem 2 and Lemma 4 to prove Theorem 1. □

**Theorem 2.** $p_{\psi_1}(X = 1) = p_{\psi_2}(X = 2) = ... = p_{\psi_d}(X = d) \iff \{\phi_j\}_{j=1}^d$ *are equiangular and equal norm.*

To show the forward ($\Rightarrow$) direction, it suffices to first show $p_{\psi_1}(X = 1) = p_{\psi_2}(X = 2) = ... = p_{\psi_d}(X = d) \Rightarrow \{\phi_j\}_{j=1}^d$ are equal norm and then show $p_{\psi_1}(X = 1) = p_{\psi_2}(X = 2) = ... = p_{\psi_d}(X = d) \Rightarrow \{\phi_j\}_{j=1}^d$ are equiangular.

**Lemma 2.** $p_{\psi_1}(X = 1) = p_{\psi_2}(X = 2) = ... = p_{\psi_d}(X = d) \Rightarrow \{\phi_j\}_{j=1}^d$ *are equal norm.*

*Proof.* This implication holds when $d = 2$:

$$p_{\psi_1}(X = 1) = \frac{e^{\|\phi_1\|}}{e^{\|\phi_1\|} + e^{\|\phi_2\| \cos(\theta_{1,2})}} = \frac{e^{\|\phi_2\|}}{e^{\|\phi_2\|} + e^{\|\phi_1\| \cos(\theta_{1,2})}} = p_{\psi_2}(X = 2)$$

$$e^{\|\phi_1\|}(e^{\|\phi_2\|} + e^{\|\phi_1\| \cos(\theta_{1,2})}) = e^{\|\phi_2\|}(e^{\|\phi_1\|} + e^{\|\phi_2\| \cos(\theta_{1,2})})$$

$$e^{\|\phi_1\| + \|\phi_1\| \cos(\theta_{1,2})} = e^{\|\phi_2\| + \|\phi_2\| \cos(\theta_{1,2})}$$

$$\|\phi_1\|(1 + \cos(\theta_{1,2})) = \|\phi_2\|(1 + \cos(\theta_{1,2}))$$

$$\|\phi_1\| = \|\phi_2\|$$

Suppose $d > 2$ and $p_{\psi_1}(X = 1) = ... = p_{\psi_d}(X = d)$. By Assumption 2, all $m-$combinations $\binom{d}{m}$ of $\{p_{\psi_1}(X = 1), ..., p_{\psi_d}(X = d)\}$ are equal. This implies all 2-combinations are equal: $p_{\psi_i}(X = i) = p_{\psi_j}(X = j) \Rightarrow \|\phi_i\| = \|\phi_j\|$. Therefore, $\|\phi_1\| = ... = \|\phi_d\|$. □

**Lemma 3.** $p_{\psi_1}(X = 1) = p_{\psi_2}(X = 2) = ... = p_{\psi_d}(X = d) \Rightarrow \{\phi_j\}_{j=1}^d$ *are equiangular.*

*Proof.* This implication is trivially true when $d = 2$ (see the proof of Lemma 2), and we show it is similarly true when $d = 3$. Following the steps in the proof of Lemma 2, we arrive at the following 3 pairs of equalities:

(1) $e^{\|\phi_1\|(1+\cos(\theta_{1,2}))} + e^{\|\phi_1\| + \|\phi_3\| \cos(\theta_{2,3})} = e^{\|\phi_2\|(1+\cos(\theta_{1,2}))} + e^{\|\phi_2\| + \|\phi_3\| \cos(\theta_{1,3})}$

(2) $e^{\|\phi_1\|(1+\cos(\theta_{1,3}))} + e^{\|\phi_1\| + \|\phi_2\| \cos(\theta_{2,3})} = e^{\|\phi_3\|(1+\cos(\theta_{1,3}))} + e^{\|\phi_3\| + \|\phi_2\| \cos(\theta_{1,3})}$

(3) $e^{\|\phi_2\|(1+\cos(\theta_{2,3}))} + e^{\|\phi_2\| + \|\phi_1\| \cos(\theta_{1,3})} = e^{\|\phi_3\|(1+\cos(\theta_{2,3}))} + e^{\|\phi_3\| + \|\phi_1\| \cos(\theta_{1,2})}$

From Lemma 2, $p_{\psi_1}(X = 1) = p_{\psi_2}(X = 2) = p_{\psi_3}(X = 3) \Rightarrow \|\phi_1\| = \|\phi_2\| = \|\phi_3\|$, so the above pairs of equalities reduce to:

$$(1) \ \cos(\theta_{2,3}) = \cos(\theta_{1,3})$$
$$(2) \ \cos(\theta_{2,3}) = \cos(\theta_{1,3})$$
$$(3) \ \cos(\theta_{1,3}) = \cos(\theta_{1,2})$$

and when $d = 3$, $\{\phi_j\}_{j=1}^3$ are equiangular.

Suppose $d > 3$ and $p_{\psi_1}(X = 1) = ... = p_{\psi_d}(X = d)$. By Assumption 2, all $m-$combinations $\binom{d}{m}$ of $\{p_{\psi_1}(X = 1), ..., p_{\psi_d}(X = d)\}$ are equal. This implies all 3-combinations are equal: $p_{\psi_i}(X = i) = p_{\psi_j}(X = j) = p_{\psi_k}(X = k) \Rightarrow \theta_{i,j} = \theta_{i,k} = \theta_{j,k}$. Therefore, all angles are equal $\theta_{i,j} = \theta_{l,m}$ for $1 \le i, j, l, m \le d$. $\qquad\square$

*Proof of Theorem 2.* ($\Rightarrow$) Suppose $p_{\psi_1}(X = 1) = p_{\psi_2}(X = 2) = ... = p_{\psi_d}(X = d)$.

By Lemma 2 and Lemma 3, $p_{\psi_1}(X = 1) = p_{\psi_2}(X = 2) = ... = p_{\psi_d}(X = d) \Rightarrow \{\phi_j\}_{j=1}^d$ are equiangular and equal norm.

($\Leftarrow$) Suppose $\{\phi_j\}_{j=1}^d$ are equiangular and equal norm. Let $\|\phi\|$ be the norm of any vector in our set and $\cos(\theta)$ be the pairwise angle between any two vectors. Then

$$p_{\psi_i}(X = i) = \frac{e^{\|\phi\|}}{e^{\|\phi\|} + (d-1)e^{\|\phi\| \cos(\theta)}} = p_{\psi_j}(X = j)$$

for any $1 \le i, j \le d$. $\qquad\square$

**Lemma 4.** *For a set of equiangular and equal norm vectors, maximum equiangularity maximizes* $\sum_j p_{\psi_j}(X = j)$.

*Proof.* The maximum pairwise angle between two vectors in $\mathbb{R}^d$ is $\pi$, and from Theorem 2

$$p_{\psi_i}(X = i) = p_{\psi_j}(X = j) = \frac{e^{\|\phi\|}}{e^{\|\phi\|} + (d-1)e^{\|\phi\| \cos(\theta)}}$$

for all $1 \le i, j \le d$. Increasing the angle $\theta$ decreases $\cos(\theta)$. Decreasing $\cos(\theta)$ only decreases the denominator, which in turn, increases $p_{\psi_i}(X = i)$. Therefore, maximizing the pairwise angle between all vectors maximizes $p_{\psi_i}(X = i)$ for all $1 \le i \le d$. $\qquad\square$

### A.2.1 AN ELMES MINIMIZES $H_{\psi_i}(X)$

*Proof of Lemma 1.* Equal norm and equiangular $\{\phi_j\}_{j=1}^d$ are bounded in norm, and thus, the set of probability distributions we obtain $\{p_{\psi_i}(1), p_{\psi_i}(2), ..., p_{\psi_i}(d)\}$ belong to a capped simplex (Warmuth & Kuzmin, 2008) $\Delta_c^d = \{p \in \Delta | \max_k p_{\psi_i}(k) \le c\}$ where $c = \frac{e^{\|\phi\|^2}}{e^{\|\phi\|^2} + (d-1)e^{\|\phi\|^2 \cos(\theta)}}$. Clearly, among such probability vectors, the minimum entropy is achieved at the boundary where $\cos(\theta)$ is minimized, i.e., when the $\{\phi_j\}_{j=1}^d$ are maximally equiangular. $\qquad\square$

### A.2.2 AN ELMES MAINTAINS PERMUTATION INVARIANCE

*Proof of Proposition 2.* This follows from row-wise equivariance to permutations in matrix multiplication. For any permutation $\pi : [1, \ldots, n] \to [1, \ldots, n]$ applied to the rows of $\mathcal{S}^n$, we have $\pi(\mathcal{S})W = \pi(\mathcal{S}W)$. □

## B EXPERIMENTAL SETTINGS

In this section, we describe our experimental settings, and further, we direct readers interested in reproducing or using any of the methods we benchmark in this work to our released code. Unless stated otherwise, all *universal meta-learning* baselines use a CLIP feature extractor to encode images.

**Large-Scale Pre-Training.** All methods evaluated in the *universal meta-learning* setting adhere to the same pre-training paradigm. For each large-scale image classification dataset, we reformulate the objective from typical supervised image classification to both a 5-way-1-shot and a 5-way-5-shot episodic prediction tasks. Within a dataset, examples from different classes are randomly sampled to compose a batch of episodes, and after exhausting iterating through every training example, this process is repeated with the next dataset. Iterating through each dataset in our set of ImageNet-1k, Fungi, MSCOCO, and WikiArt then constitutes a single epoch of training.

**ProtoNet and MetaOpt Implementations.** For the ProtoNet and MetaOpt algorithms, we evaluate two settings. The first freezes the CLIP backbone and then applies the metric-learning objective— cosine distance for ProtoNet and SVM for MetaOpt—to classify the query image from the unmodified CLIP embeddings. The second emulates P>M>F Hu et al. (2022) by fine-tuning the CLIP backbone during large-scale pre-training with the metric-learning objective function. During inference, the metric-learning objective is applied to the fine-tuned CLIP embeddings to classify query images.

**MetaQDA Implementation.** We follow the MetaQDA algorithm presented in Zhang et al. (2021). Specifically, we freeze the CLIP feature extractor backbone and train the MetaQDA classifier during large-scale episodic pre-training.

**SNAIL Implementation.** We use the architecture presented in Mishra et al. (2017) but with the hidden dimension of the Attention and Temporal Convolution Blocks adapted to CLIP embeddings rather than the ResNet embeddings used in the original implementation. As in this Mishra et al. (2017), we freeze the CLIP feature extractor and train the SNAIL model parameters during large-scale pre-training.

**GPICL Implementation.** We adapt the GPICL algorithm presented by Kirsch et al. (2022) for episodic meta-training with an ELMES label encoder. Specifically, we represent image feature vectors as CLIP embeddings and the label embeddings with an ELMES. Following Kirsch et al. (2022), we form a sequence by concatening the current CLIP image embedding *with the previous* example's ELMES label embedding and add learnable positional embeddings so the model can use positional information of elements in the sequence to classify the query point in a causal-like fashion. We set the General-Purpose In-Context Learning Transformer model to a ViT-Large (Dosovitskiy et al., 2020) with leranable positional embeddings.

**CAML Implementation.** The image encoder is set to CLIP and the label encoder is an ELMES. For the non-causal sequence model, we use a ViT-Large as described in Table 1 of Dosovitskiy et al. (2020). This size is chosen as it has a hidden dimension of 1,024 and the CLIP output embedding vectors have hidden dimension of size 768. Choosing a non-causal sequence model with a large hidden dimension allows us to concatenate the label embedding to the CLIP embedding; in this case, the label embedding is a 256 dimensional ELMES. In total, the implementation of CAML used for empirical evaluation has 302 million trainable parameters.

**Optimization Settings.** Following the recommendation of training Vision Transformers (Steiner et al., 2021) as well as standard practices, all universal meta-learning approaches use a cosine learning rate schedule with 9,600 warmup steps increasing linearly from 0 to 1e-5 followed by cosine decay to 1e−6 over the subsequent 360,000 steps. Given the size of our pre-training datasets, we do not apply dropout, attention dropout, or weight decay regularization. We select a batch size of 525 so the 5-way-1-shot episodes contain 520 query predictions and the 5-way-5-shot episodes contain 500

Table 5: **MiniImageNet & CIFAR-fs** mean accuracy and standard error across 10,000 test epochs.

| Method | CIFAR-fs | | MiniImageNet | |
|---|---|---|---|---|
| | 5w-1s | 5w-5s | 5w-1s | 5w-5s |
| CAML [ELMES Class Embedding] | **70.8**±.2 | **85.5**±.1 | **96.2**±.1 | **98.6**±.0 |
| CAML [Learnable Class Embedding] | **71.1**±.2 | **85.9**±.1 | **96.1**±.1 | **98.7**±.0 |

Table 6: **CUB & tiered-ImageNet & Aircraft** mean accuracy and standard error across 10,000 test epochs.

| Method | CUB | | tiered-ImageNet | | Aircraft | |
|---|---|---|---|---|---|---|
| | 5w-1s | 5w-5s | 5w-1s | 5w-5s | 5w-1s | 5w-5s |
| CAML [ELMES Class Embedding] | **91.8**±.2 | **97.1**±.1 | **95.4**±.1 | 98.1±.1 | 63.3±.3 | 79.1±.2 |
| CAML [Learnable Class Embedding] | **91.8**±.2 | **97.1**±.1 | 95.3±.1 | **98.3**±.1 | **66.3**±.2 | **80.6**±.2 |

Table 7: **Pascal & Paintings** mean accuracy and standard error across 10,000 test epochs.

| Method | Pascal + Paintings | | Paintings | | Pascal | |
|---|---|---|---|---|---|---|
| | 5w-1s | 5w-5s | 5w-1s | 5w-5s | 5w-1s | 5w-5s |
| CAML [ELMES Class Embedding] | **63.8**±.2 | **78.3**±.1 | 51.1±.2 | **65.2**±.1 | **82.6**±.2 | **89.7**±.1 |
| CAML [Learnable Class Embedding] | 63.1±.2 | 78.0±.1 | **51.3**±.2 | 65.0±.1 | 82.1±.2 | **89.7**±.1 |

Table 8: **meta-iNat & tiered meta-iNat & ChestX** mean accuracy and standard error across 10,000 test epochs.

| Method | meta-iNat | | tiered meta-iNat | | ChestX | |
|---|---|---|---|---|---|---|
| | 5w-1s | 5w-5s | 5w-1s | 5w-5s | 5w-1s | 5w-5s |
| CAML [ELMES Class Embedding] | 91.2±.2 | 96.3±.1 | 81.9±.2 | 91.6±.1 | **21.5**±.1 | 22.2±.1 |
| CAML [Learnable Class Embedding] | **91.4**±.2 | **96.4**±.1 | **82.1**±.2 | **91.8**±.1 | **21.5**±.1 | **22.6**±.1 |

query predictions. Given the scale of the pre-training datasets—and the computation to train a single model—we do not conduct any hyperparameter tuning.

**P>M>F Meta-Training.** We follow the settings used by Hu et al. (2022) to evaluate P>M>F. Specifically, P>M>F uses a DINO (Caron et al., 2021) feature extractor rather than a CLIP feature extractor as the authors of P>M>F found a DINO feature extractor to be preferrable. We refer readers Hu et al. (2022) for this comparison. For meta-training, we use the code released by Hu et al. (2022) and simply switch out the datasets to evaluate the In-Domain setting. Both the in-domain and universal meta-learning settings use the same test-set data; the difference is that P>M>F meta-trains on each training dataset before evaluating on the testing dataset of each benchmark.

## C  SUPPLEMENTARY ANALYSIS

**ELMES Ablation.** To supplement our theoretical analysis in Section 4, we train a version of CAML with learnable class embedding vectors in place of the fixed ELMES encoder. Given our analysis in Section 4, it is perhaps unsurprising we find that—without any constraints or limitations—the class embeddings converge to an ELMES. The average pair-wise angle between embedding vectors is $1.77 \pm 0.02$ radians whereas the expected pairwise angle from an ELMES is $1.82$. Similarly, the average norm of the learnable class embeddings converges to $1.34 \pm 0.02$ whereas the learned norm of the ELMES model is $1.32$.

An evaluation comparing CAML with learnable class embeddings to the approach with a fixed ELMES encoder is presented in Table 5, Table 6, Table 7, and Table 8 of the Appendix. In summary, the performance is approximately the same on each benchmark with the exception of Aircraft. In this case, the learnable embedding model significantly outperforms the ELMES model, and moreover, surpasses all other universal meta-learning baselines on the 5-way-1-shot split with an accuracy of $66.3 \pm .2$. Nevertheless, given the similarity between both approaches on the remaining 10 datasets,

Table 9: **MiniImageNet & CIFAR-fs** mean accuracy and standard error across 10,000 test epochs. ◦ indicates mean and standard error across 2,500 test epochs.

| Method | CIFAR-fs | | MiniImageNet | |
|---|---|---|---|---|
| | 5w-1s | 5w-5s | 5w-1s | 5w-5s |
| CAML (ResNet34) | 61.8 ± .2 | 79.4 ± .2 | 94.7 ± .1 | 98.1 ± .0 |
| CAML (ViT-base) | 70.8±.2 | 85.5±.1 | 96.2±.1 | 98.6±.0 |
| CAML (ViT-huge)° | **83.3±.4** | **93.5±.2** | **98.6±.1** | **99.6±.0** |

Table 10: **CUB & tiered-ImageNet & Aircraft** mean accuracy and standard error across 10,000 test epochs. ◦ indicates mean and standard error across 2,500 test epochs.

| Method | CUB | | tiered-ImageNet | | Aircraft | |
|---|---|---|---|---|---|---|
| | 5w-1s | 5w-5s | 5w-1s | 5w-5s | 5w-1s | 5w-5s |
| CAML (ResNet34) | 75.4 ± .2 | 88.3 ± .1 | 96.1 ± .1 | 98.5 ± .0 | 45.1 ± .2 | 58.7 ± .2 |
| CAML (ViT-base) | 91.8±.2 | 97.1±.1 | 95.4±.1 | 98.1±.1 | 63.3±.3 | 79.1±.2 |
| CAML (ViT-huge)° | **95.8±.2** | **98.7±.1** | **96.8±.2** | **98.8±.1** | **81.8±.4** | **92.1±.3** |

Table 11: **Pascal & Paintings** mean accuracy and standard error across 10,000 test epochs. ◦ indicates mean and standard error across 2,500 test epochs.

| Method | Pascal + Paintings | | Paintings | | Pascal | |
|---|---|---|---|---|---|---|
| | 5w-1s | 5w-5s | 5w-1s | 5w-5s | 5w-1s | 5w-5s |
| CAML (ResNet34) | 57.5 ± .2 | 71.0 ± .1 | 46.1 ± .2 | 57.3 ± .1 | 77.4 ± .2 | 86.8 ± .1 |
| CAML (ViT-base) | 63.8±.2 | 78.3±.1 | 51.1±.2 | 65.2±.1 | 82.6±.2 | 89.7±.1 |
| CAML (ViT-huge)° | **66.4±.4** | **81.0±.2** | **54.7±.3** | **69.9±.2** | **83.4±.4** | **90.1±.3** |

Table 12: **meta-iNat & tiered meta-iNat & ChestX** mean accuracy and standard error across 10,000 test epochs. ◦ indicates mean and standard error across 2,500 test epochs.

| Method | meta-iNat | | tiered meta-iNat | | ChestX | |
|---|---|---|---|---|---|---|
| | 5w-1s | 5w-5s | 5w-1s | 5w-5s | 5w-1s | 5w-5s |
| CAML (ResNet34) | 82.4 ± .2 | 91.4 ± .1 | 72.3 ± .2 | 84.6 ± .2 | **21.8 ± .1** | **23.6 ± .1** |
| CAML (ViT-base) | 91.2±.2 | 96.3±.1 | 81.9±.2 | 91.6±.1 | 21.5±.1 | 22.2±.1 |
| CAML (ViT-huge)° | **94.6±.3** | **97.9±.1** | **89.3±.4** | **95.6±.2** | 21.6±.2 | 22.0±.2 |

and the learnable class embeddings actually forming an ELMES, we attribute the difference in Aircraft performance to stochasticity in training the model, suggesting that the fixed ELMES encoder is indeed optimal.

**Image Encoder Ablation.** To evaluate how the performance of CAML is affected by the pre-trained image encoder, we evaluate CAML with a ResNet-34 image encoder pre-trained on ImageNet-1k, a ViT-base image encoder pre-trained with CLIP, and a ViT-huge image encoder that is pre-trained on Laion-2b (Schuhmann et al., 2022). We use the open source models released by Hugging Face in our evaluation.

As indicated in Table 9, Table 10, Table 11, and Table 12, the performance of CAML scales with the strength of the feature extractor. Specifically, the performance with a ResNet-34 feature extractor is significantly worse than the performance with a CLIP ViT-base feature extractor, and in turn, the performance with a CLIP ViT-base is significantly worse than the performance with a Laion-2b ViT-huge feature extractor. However, its unclear what facet of the improved feature extractor is relevant for CAML , especially on out-of-distribution tasks like Aircraft where the most benefit is seen. Moreover, it is unclear why there is no improvement on another out-of-distribution dataset, ChestX.

**t-SNE Plots of Image Encoder Embeddings**

Figure 4: t-SNE projections of different image embeddings of various benchmark datasets with embeddings colored class identity. We see ViT-huge trained with Laion-2b better separates the Aircraft dataset than does ViT-base trained with CLIP. However, both image encoders are unable to separate ChestX.

To investigate this dimension, we visualize the image embeddings of both Aircraft and ChestX using t-sne (Van der Maaten & Hinton, 2008) dimensionality reduction. Figure 4 visualizes these embeddings colored by class identity. We find the ViT-huge model pre-trained on Laion-2b better separates the Aircraft dataset than the ViT-base model pre-trained using the CLIP objective; however, both models do not reasonably separate ChestX. We postulate that an image encoder that can capture the axes of variability among image embeddings is crucial for strong CAML performance, and the reason we observe significantly improved results on Aircraft but not ChestX when using a Laion-2b ViT-h image encoder.

Taken together, these results indicate CAML is modular: as foundational model feature extractors continue to improve, CAML will be able to capture these advances to improve its own performance.

**Assignment of Labels to Support Set Classes Analysis.** Symmetry to the assignment of labels to support set classes is a desirable property of few-shot learning algorithms. For instance, the predictions for [(bear, 1), (tower, 2), (tree, 3)] should be the same if the labels are permuted to [(bear, 3), (tower 1), (tree, 2)]. CAML is not invariant to permutations in the assignment of classes to support set examples as implied by eq. (1) in Section 4.2; however, we empirically find it is robust to them. Label symmetry is distinct from the permutation invariance property of CAML that is discussed in Section 4.3. Tangibly for the sequence [(bear, 1), (tower, 2), (tree, 3)], permutation invariance ensures the predictions are the same as if the order of demonstrations is permuted to [(tower, 2), (tree, 3), (bear, 1)].

In Figure 5(left), we visualize the histogram of the correct class probability for the example presented in Figure 2a after permuting the assignment of labels to support-set images for all 120 permutations

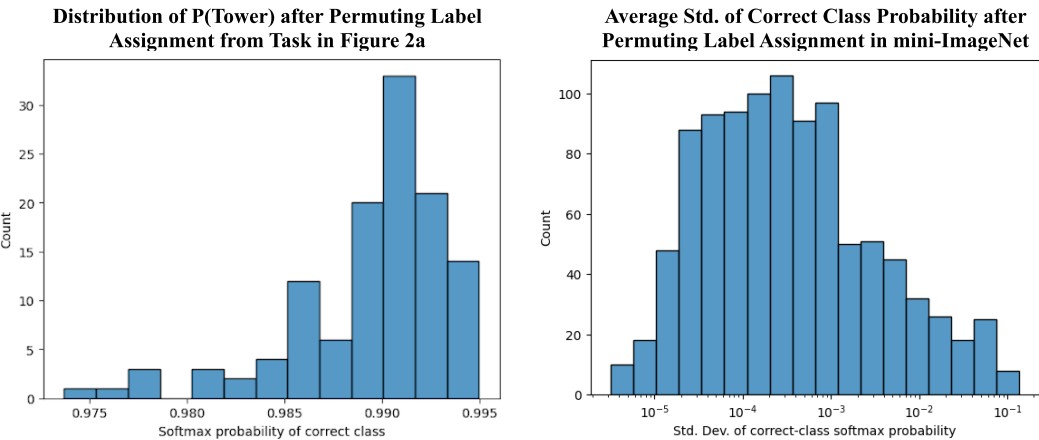

Figure 5: (Left) histogram of the correct class probability for the example presented in Figure 2a after permuting the assignment of labels to support-set images for all 120 permutations of the 5-way-1-shot task. (Right) histogram of the average standard deviation of all 120 permutations of the 5-way-1-shot task for 1,000 samples from mini-ImageNet.

of the 5-way-1-shot task. In Figure 5(right), we visualize the average standard deviation of all 120 permutations of the 5-way-1-shot task for 1,000 samples from mini-ImageNet. The mean of this statistic is $0.004 \pm 0.0004$. Taken together, this indicates CAML is empirically robust to permutations in the assignment of labels to support set classes.

## D    DISCUSSION

**Weaknesses of CAML.** Despite its strong empirical performance, CAML presents several weaknesses. First, the maximum number of classes present in the support set at any point during inference must be known at pre-training to instantiate a $d$-way ELMES. Further, at least one dataset during pre-training must use a $d$-way classification setting so the $\psi_i$ class detectors referenced in Section 4 are trained within the Transformer encoder's attention layers.

**Why does CAML not fine-tune the image encoder during pre-training?** We do not fine-tune the image encoder because it is not advantageous for *universal meta-learning*.

Our goal is to develop a meta-learning algorithm that may function in a ChatGPT-like application; it should be able to run in-context learning on any set of images. Foundational image models are trained for exactly this purpose: they are pre-trained on billions of images to form a well-structured image embedding space that is robust to augmentations, occlusions, etc. Moreover, valuable characteristics such as the presence of objects, textures, etc. of an image are encoded into the structure of the embedding space so that the axes of variability among the embeddings encode variation in specific visual attributes.

Fine-tuning the image encoder can corrupt this embedding space; especially since the datasets we use for pre-training are orders of magnitude smaller than the ones used to train the Foundational model. This hypothesis is supported by our experiments with ProtoNet and MetaOpt in Tables 1 to 4. Specifically, we find fine-tuning the backbone during pre-training leads to performance degradation on many of our benchmarks when evaluated in the universal meta-learning setting.

