# OpenReview forum: "Context-Aware Meta-Learning"
_ICLR.cc/2024/Conference — ICLR 2024 poster_

### Official Review · Reviewer_Hb5r · 2023-10-30

**Soundness:** 3 good
**Presentation:** 3 good
**Contribution:** 2 fair
**Rating:** 6
**Confidence:** 3

**Summary:**

This work introduces a novel meta-learning algorithm that allows visual models to learn new concepts during inference, mimicking the capabilities of LLMs such as ChatGPT. The technique utilizes a static pre-trained feature extractor and treats meta-learning similarly to sequence modeling with labeled and unlabeled data points. When evaluated on 11 benchmarks, the proposed method, without any meta-training or fine-tuning, outperformed or matched the leading P>M>F algorithm in 8 of those benchmarks.

**Strengths:**

**Intriguing Research Question:** This paper delves into a significant question in meta-learning. The authors note that meta-learning traditionally involves pretraining, meta-learning, and fine-tuning. However, their approach seeks to bypass meta-learning and fine-tuning by transforming the learning process into a sequence modeling task and applying the in-context learning objective function.

**Rigorous Numerical Performance:** The paper's data shows the model's performance to be both impressive and robust.

**Novel ELMES Class Encoder:** The class embedding presented here appears to be quite innovative. I've noticed no other usage of this type of class embedding in in-context learning in NLP or VLM literature unless other reviewers bring up some status quo. This is also in alignment with some recent studies on Arxiv that indicate that class embeddings might be unnecessary, as arbitrary words, numbers, or first names can be used to label images (see [2]).

**Weaknesses:**

I will provide the paper's weaknesses in the following.
- The concept of using the ICL objective function to pre-train a transformer model isn't a new one [1]. Unlike [1], which pre-trained a transformer from scratch within a meta-learning framework, this paper adapts the pre-training objective to Clip image embeddings, which doesn't significantly enhance novelty.
- The primary innovative aspect highlighted in this paper is the ELMES Class Encoder. While this feature is intriguing, it narrows the scope of innovation in the study.
- Some claims in the paper lack experimental backing. For instance, the assertion that the ELMES Class Encoder upholds label symmetry and is invariant to the permutation of demonstrations isn't convincingly proven with data.
- The mathematical explanations in Section 4 are challenging to follow. A clearer, more comprehensible presentation of this section would be beneficial. Until then, I'm relying on other reviewers to verify the accuracy of the mathematical derivations presented.

**Questions:**

- Could the authors elaborate on what they consider to be the primary innovative contribution of their study?
- It would be beneficial if the authors could include experiments to demonstrate the label symmetry and permutation invariance capabilities of the demonstrations as claimed.

[1]. General-Purpose In-Context Learning by Meta-Learning Transformers, NeurIPs 2022

[2]. Small Visual Language Models can also be Open-Ended Few-Shot Learners, Arxiv 2023.

---

> ### Author Response · Authors · 2023-11-17
> **Response to Reviewer Hb5r**
>
> Thank you for reviewing our work! We appreciate you noting that our work addresses an intriguing research question, presents a rigorous empirical performance, and offers an innovative class embedding with the ELMES label encoder. We’re similarly excited for future explorations into class embeddings, and hope our theoretical analysis might be beneficial to others working in this space. Addressing your questions:
>
> >1. The concept of using the ICL objective function to pre-train a transformer model isn't a new one [1]. Unlike [1], which pre-trained a transformer from scratch within a meta-learning framework, this paper adapts the pre-training objective to Clip image embeddings, which doesn't significantly enhance novelty.
>
> GPICL uses a causal approach by formulating the input as a sequence of (image, previous image’s label) pairs. Specifically, GPICL must learn to look at previous elements in the sequence to associate a label with the current element’s image and relies on positional encodings to keep track of where different elements are in the sequence.
>
> Similar to GPICL, we initially tried a causal approach; however, we couldn’t get this to work and it also conflicted with our intuition that permutation invariance—and therefore non-causal learning—would be an important property for meta-learning applications. This understanding led us to CAML as causal learning mechanisms affect empirical performance. Tangibly in Section 5.2, we observe an average difference of 7.3 accuracy points between CAML and GPICL—even when GPICL uses a CLIP encoder and ELMES label encoder—and hypothesize the causal approach actually forces a subtly more difficult modeling problem by imposing a causal inductive bias on a fundamentally non-causal prediction task.
>
> **GPICL Implementation** We adapt the GPICL algorithm presented by [1] for episodic meta-training with an *ELMES* label encoder. Specifically, we represent image feature vectors as CLIP embeddings and the label embeddings with an *ELMES*. Following [1], we form a sequence by concatening the current CLIP image embedding *with the previous* example's *ELMES* label embedding and add learnable positional embeddings so the model can use positional information of elements in the sequence to classify the query point in a causal-like fashion. We set the General-Purpose In-Context Learning Transformer model to a ViT-Large with leranable positional embeddings.
>
> | Approach | (5w-1s) CIFAR-fs | (5w-1s) MiniImageNet | (5w-1s) Pascal + Paintings | (5w-1s) Paintings | (5w-1s) Pascal |
> | :-----: | :-----: | :-----: | :-----: | :-----: | :-----: |
> |GPICL| 41.5 | 95.6 | 62.6 | **51.6** | 81.7 |
> | CAML | **70.8** | **96.2** | **63.8** | 51.1 | **82.6** |
>
> | Approach | (5w-1s) meta-iNat | (5w-1s) tiered meta-iNat | (5w-1s) ChestX | (5w-1s) CUB | (5w-1s) tiered-ImageNet | (5w-1s) Aircraft |
> | :-----: | :-----: | :-----: | :-----: | :-----: | :-----: | :-----: |
> |GPICL | 90.0 | 60.8 | 20.1 | 75.1 | 94.6 | 19.8 |
> | CAML | **91.2** | **81.9** | **21.5** | **91.8** | **95.4** | **63.3**|
>
> | Approach | (5w-5s) CIFAR-fs | (5w-5s) MiniImageNet | (5w-5s) Pascal + Paintings | (5w-5s) Paintings | (5w-5s) Pascal |
> | :-----: | :-----: | :-----: | :-----: | :-----: | :-----: |
> |GPICL | 78.3 | 98.2 | 74.6 | 61.0 | 88.2 |
> |CAML | **85.5** | **98.6** | **78.3** | **65.2** | **89.7**|
>
> | Approach | (5w-5s) meta-iNat | (5w-5s) tiered meta-iNat | (5w-5s) ChestX | (5w-5s) CUB | (5w-5s) tiered-ImageNet | (5w-5s) Aircraft |
> | :-----: | :-----: | :-----: | :-----: | :-----: | :-----: | :-----: |
> |GPICL | 95.1 | 87.6 | 20.9 | 94.5 | 97.2 | 61.8 |
> | CAML | **96.3** | **91.6** | **22.2** | **97.1** | **98.1** | **79.1** |

---

> ### Author Response · Authors · 2023-11-17
> **Response to Reviewer Hb5r [Part II]**
>
> >2. Some claims in the paper lack experimental backing. For instance, the assertion that the ELMES Class Encoder upholds label symmetry and is invariant to the permutation of demonstrations isn't convincingly proven with data.
>
> This is a great question; we decompose our response into two parts:
>
> *1. ELMES Class Encoder Permutation Invariance w.r.t. demonstration order.*
>
> Suppose we have a sequence of (image, label) pairs: [(bear, 1), (tower, 2), (tree, 3)]. Then Permutation Invariance w.r.t. demonstration order means the model's predictions should be the same if we change the order to [(tree, 3), (tower, 2), (bear, 1)]. **In particular, this property is not conveyed by [1,2] as they rely on the order of demonstrations to make predictions.** We prove CAML is permutation invariant in Lemma 2, and we empirically check our proof by examining the logits of all 120 permutations of the example presented in Figure 2(a) and verify that there is no variation.
>
> *2. Meta-Learning label symmetry.*
>
> Label symmetry is a property of few-shot classification. For instance, the predictions for [(bear, 1), (tower, 2), (tree, 3)] should be the same if the labels are permuted to [(bear, 3), (tower 1), (tree, 2)]. CAML is not invariant to permutations in the assignment of classes to support set examples as implied by Equation (1) in Section 4.2; however, we empirically find it is robust to them. Specifically, we find that CAML is approximately invariant to permutations in the assignment of labels to support set classes; the average standard deviation in the correct class prediction probability is only 0.004 on mini-ImageNet. We expand on this point with additional analysis in the updated manuscript in Section C of the Appendix under header Assignment of Labels to Support Set Classes Analysis and Figure 5.
>
> >3. Unclear Mathematical Explanation.
>
> Due to space constraints, we relocated a significant amount of theoretical exposition to Section A.1.1 of the supplementary material. We can work towards elucidating Section 4; however, the material spans Geometry (regular simplexes inscribed within a sphere), Information Theory (entropy, optimal coding), Representation Theory (Equiangular Tight Frames, Welch Lower Bound), and Group Theory (SO(3) ELMES invariance). In the meantime, we hope that the theoretical analysis in Section 4, if imperfect, may still be beneficial to readers in understanding the theoretical motivation of CAML.

---

> ### Author Response · Authors · 2023-11-17
> **Response to Reviewer Hb5r [Part III]**
>
> >4. Could the authors elaborate on what they consider to be the primary innovative contribution of their study?
>
> Thank you for posing this question. Based on your review, it’s clear that our contribution was not clearly communicated in our initial submission. We think this is a critical point—and specifically address your question in our response to all reviewers—but we’d like to extend that response here as well.
>
> Our primary contribution is twofold.
>
> **1. We develop a new meta-learning evaluation paradigm (universal meta-learning) that approximates the performance of visual meta-learning algorithms in a ChatGPT-like application.**
>
> That is, we want to see how visual meta-learners perform on diverse tasks unseen at training, with no parameter updates. This is hard: we want a model that will work well for any set of images, but actually deploying to users and quantifying failure cases is impractical for many research groups. Our best proxy to measure a model’s capacity to generalize in a ChatGPT-like application is to evaluate it on a diverse set of meta-learning benchmarks spanning many different image classification paradigms without meta-training on any of their training sets or fine-tuning on the support set during inference.
>
> *What is the motivation?*
>
> We’d like to develop an evaluation paradigm for meta-learning models that quantifies how they would perform in a ChatGPT-like application. Specifically, how do meta-learning algorithms perform when a user can upload any set of images and serving-time constraints prevents fine-tuning on the support set?
>
> *What is missing in current meta-learning evaluation paradigms?*
>
>  In-domain meta-learning paradigms measure generalization to unseen classes during inference by training on closely related classes during training (i.e. meta-training). Cross-domain meta-learning refers to a challenging evaluation paradigm where the meta-training and inference-time data distributions are significantly different, and the goal is to maximize the performance on the inference-time data distribution by fine-tuning on the support set.
>
> There always seems to be a one-to-one or many-to-one correspondence between the set of classes used for training and the set of classes used during inference. A sense of many-to-many, or simply “do well on everything”, isn’t captured by either in-domain or cross-domain evaluation paradigms, and even for cross-domain settings, the performance on classes similar to those used for meta-training is often not evaluated.
>
> *What does setting universal meta-learning specifically address?*
>
> Universal meta-learning specifically addresses a serving-time application of few-shot image classification where the user can upload any set of images and only specify which images are similar (i.e. belong to the same class). Meta-training on a set of related images is impossible, because the space of all possible images that could be uploaded by the user can’t be covered. Moreover, fine-tuning on the support set is also not feasible for inference-time applications deployed in a ChatGPT-like setting.
>
> We define the universal meta-learning evaluation setting to measure progress in this area. This type of evaluation paradigm already exists for NLP—and is quite popular—but we are unfamiliar with one if it already exists for few-shot image classification.

---

> ### Author Response · Authors · 2023-11-17
> **Response to Reviewer Hb5r [Part IV]**
>
> **2. We develop a meta-learning algorithm that works well in this setting (i.e. CAML).**
>
> CAML is a sequence-based meta-learning algorithm that is designed to generalize to any set of images during inference without fine-tuning (thereby addressing the problem). In our experiments, we find that CAML surpasses the performance of other meta-learning algorithms in the *universal meta-learning* evaluation setting. Surprisingly, its performance in the *universal meta-learning* setting often matches–and even exceeds—the in-domain meta-learning performance of the state-of-the-art meta-learning algorithm, P>M>F, that is directly meta-trained on each benchmark.
>
> CAML is designed to perform well in the *universal meta-learning* setting – that is, it’s “general and fast” – while prior approaches are not. Specifically, and distinct from prior sequence-based meta-learning algorithms like GPICL [1] and SNAIL [2],
>
> 1. CAML is invariant to permutations in the support set and non-causal. In our experiments, we observe a performance gap between both approaches and CAML and hypothesize the causal approach actually forces a subtly more difficult modeling problem by imposing a causality inductive bias on a fundamentally non-causal prediction task.
> 2. CAML uses an Equal Length and Maximally Equiangular Set (ELMES) to integrate the label information from the support set into the model. Our theoretical analysis shows this formulation minimizes the entropy of identifying class labels.
> 3. CAML represents an (image, label) element of the support set as a single vector that contains information about both the image and the label. Specifically, a Foundational Model Image Encoder embeds the support set image, an ELMES embeds the label, and both vectors are concatenated to form a single demonstration vector from the (image, label) pair in the support set.
>
> **These changes make a big difference. In our experiments,  we observe a performance gap of (on average) 5.2 accuracy points between CAML and SNAIL and 7.3 accuracy points between CAML and GPICL for each benchmark.**
>
> ---
>
> [1] General-Purpose In-Context Learning by Meta-Learning Transformers, NeurIPs 2022
>
> [2] Mishra, Nikhil, Mostafa Rohaninejad, Xi Chen, and Pieter Abbeel. "A simple neural attentive meta-learner.", ICLR 2018

---

> > ### Author Response · Authors · 2023-11-22
> > **Follow-up**
> >
> > Dear Reviewer Hb5r,
> >
> > Thank you again for reviewing our work! As the discussion period ends shortly, we wanted to check if you have any further questions or found our responses helpful?
> >
> > Please let us know and thank you for your time!

---

> > > ### Comment · Reviewer_Hb5r · 2023-11-23
> > >
> > > I appreciate the authors' comprehensive responses to my inquiries and those from other reviewers. I have carefully reviewed all of their replies. The authors' clarifications have effectively highlighted their work's novelties and contributions, which I find satisfactory. Additionally, their comparison of the proposed method with GPICL and SNAIL is noteworthy, and the demonstrated improvements are enough. Given the authors' satisfactory answers, I am inclined to raise my current score to 6.

---

### Official Review · Reviewer_yxC1 · 2023-10-30

**Soundness:** 3 good
**Presentation:** 3 good
**Contribution:** 3 good
**Rating:** 6
**Confidence:** 3

**Summary:**

The paper proposes a meta-learning algorithm that learns new visual concepts during inference without fine-tuning. The method performs well on several benchmark datasets.

**Strengths:**

1. The paper draws a new perspective for meta-learning: learning to classify a query from a context of support set, imitating the way in LLMs.
2. The framework is straightforward and clean. The reason (proved theoretically) for using the specific ELMES embedding is presented well.
3. Extensive experiments and analysis are provided.

**Weaknesses:**

Apologies in advance I am not an expert in meta-learning. But I still have the following questions:
1. What is the unknown class embedding initialization for the ELMES Class Encoder?
2. As discussed by the authors themselves in section 5.3, the number of classes need to be known in advance and the frozen encoders limit the learning ability. However in my view, the need of number of classes is an inherent problem in few-shot learning. But finetuning more modules could be further discussed.
3. There is a lack of experimental details (especially training details).

**Questions:**

See Weaknesses.

**Details Of Ethics Concerns:**

None.

---

> ### Author Response · Authors · 2023-11-17
> **Response to Reviewer yxC1**
>
> Thank you for reviewing our work! We appreciate you noting the extensiveness of our experiments and analysis as well as the elegance of our proposed approach. Addressing your questions:
>
> >1. What is the unknown class embedding initialization for the ELMES Class Encoder?
>
> Similar to the cls token in ViT image classification, we initialize the unknown class embedding using the default torch.nn.Parameter initialization and learn this value during large-scale pre-training. We have updated the manuscript to emphasize this point.
>
> >2. Fine-tuning more modules could be discussed.
>
> We do not fine-tune the image encoder because it is not optimal for *universal meta-learning*.
>
> Our goal is to develop a meta-learning algorithm that may function in a ChatGPT-like application; it should be able to run in-context learning on any set of images. Foundational image models are trained for exactly this purpose: they are pre-trained on billions of images to form a well-structured image embedding space that is robust to augmentations, occlusions, etc. Moreover, valuable characteristics such as the presence of objects, textures, etc. of an image are encoded into the structure of the embedding space so that the axes of variability among the embeddings encode variation in specific visual attributes.
>
> Fine-tuning the image encoder can corrupt this embedding space; especially since the datasets we use for pre-training are orders of magnitude smaller than the ones used to train the Foundational model. This hypothesis is supported by our experiments with ProtoNet and MetaOpt in Tables 1, 2, 3, and 4. Specifically, we find fine-tuning the backbone during pre-training leads to performance degradation on many of our benchmarks when evaluated in the *universal meta-learning* setting.
>
> Thanks to your comment—and realizing this discussion is missing from the manuscript—we have updated the paper to include this discussion in Section D of the Appendix.
>
> >3. There is a lack of experimental details (especially training details).
>
> Thank you for pointing out this weakness; these details were sorely missing from the initial submission. We have updated the manuscript to describe the experimental settings in-depth in Section B of the Appendix. Please let us know if anything is unclear. Our code is also released in the supplementary material download attached to this submission and we commit to releasing a github repository after the review process to facilitate reproducing our results.

---

> > ### Author Response · Authors · 2023-11-22
> > **Follow-up**
> >
> > Dear Reviewer yxC1,
> >
> > Thank you again for reviewing our work! As the discussion period ends shortly, we wanted to check if you have any further questions or found our responses helpful?
> >
> > Please let us know and thank you for your time!

---

> > > ### Comment · Reviewer_yxC1 · 2023-11-22
> > > **Sorry for the late reply and please be careful**
> > >
> > > Thank the authors for their replies and my concerns are resolved. Apologies for the late reply.
> > >
> > > I appreciate the effort of the authors for revising the paper. HOWEVER, the page limit of the revised version shall NOT exceed the page limit of 9. Please be careful.

---

> > > > ### Author Response · Authors · 2023-11-22
> > > >
> > > > Thanks for catching this! We've updated the manuscript so the revised version is now at 9 pages.
> > > >
> > > > Sincerely,
> > > >
> > > > Submission8213 authors

---

> > > > > ### Comment · Reviewer_yxC1 · 2023-11-22
> > > > > **Update**
> > > > >
> > > > > Thank the authors for your timely response! I appreciate the efforts made and I am keeping my score as 6.

---

### Official Review · Reviewer_GdKg · 2023-10-31

**Soundness:** 3 good
**Presentation:** 3 good
**Contribution:** 1 poor
**Rating:** 5
**Confidence:** 3

**Summary:**

The paper introduces a new “universal meta-learning“ setup that “avoids meta-training on the train/validation splits of meta-learning benchmarks or fine-tuning on the support set during inference.” Instead, the paper attempts to recast meta-learning as a sequence modelling problem, where meta-testing on new tasks is analogous to in-context learning in large language models. The proposed approach CAML, context-aware meta-learning, uses CLIP image representations, together with one-hot label encoding dubbed as “Equal Length and Maximally Equiangular Set (ELMES) encoding,”  to represent each in-context learning example. The base sequence model is a Transformer encoder. It is trained to predict the query class label given in-context examples that are comprised of labelled support examples and a query example. The model is pre-trained on ImageNet-1k, Fungi, MSCOCO, and WikiArt and evaluated on 11 meta-learning benchmark datasets. Empirical results show that the proposed approach outperforms other “universal meta-learning“ baselines on 15 of 22 evaluation settings.

**Strengths:**

1. The paper is well-motivated on the need for in-context learning by drawing analogies with large language models. I also liked the analysis in Fig. 2, which illustrates how dynamic in-context examples impact representation learning

2. Theoretical analysis of the “Equal Length and Maximally Equiangular Set (ELMES) encoding” presents an interesting analysis of label symmetry and permutation invariance in meta-learning.

3. The paper presents competitive empirical results on various meta-learning baselines.

**Weaknesses:**

1. Novelty: the paper does not discuss previous work that also formulated meta-learning as a sequence, or set, modelling problem [1, 2]. The problem formulations in [1, 2] are highly similar to the proposal in this paper, except for architectural differences in implementation. This weakens the novelty of this paper.

2. The dichotomy between “meta-training“ and ”universal meta-learning”: the paper attempts to make the distinction between ”universal meta-learning” and "meta-training" in that the proposed CAML approach does not perform “meta-training“ or “fine-tuning on the support set.“ Instead, ”universal meta-learning” only performs pre-training. However, I think this dichotomy is not well-defined. Pre-training in the CAML fashion can be understood as learning across many different tasks in the pre-training dataset, i.e., meta-training, and in-context learning can be understood as performing implicit gradient descent based on the in-context examples. This dichotomy also implicates the comparisons in empirical results as CAML was mostly compared with other “universal meta-learning“ methods, i.e., ProtoNet, MetaOpt, and MetaQDA.

Additional related work:

[1] Mishra, Nikhil, Mostafa Rohaninejad, Xi Chen, and Pieter Abbeel. "A simple neural attentive meta-learner." *arXiv preprint arXiv:1707.03141* (2017).

[2] Lee, J., Lee, Y., Kim, J., Kosiorek, A., Choi, S. &amp; Teh, Y.W.. (2019). Set Transformer: A Framework for Attention-based Permutation-Invariant Neural Networks. <i>Proceedings of the 36th International Conference on Machine Learning</i>, in <i>Proceedings of Machine Learning Research</i> 97:3744-3753 Available from https://proceedings.mlr.press/v97/lee19d.html.

**Questions:**

1. Please summarize the novelty of this paper in relation to [1, 2].

2. Please respond to Weakness 2: The dichotomy between “meta-training“ and ”universal meta-learning.”

3. Please elaborate on how the transformer encoder is implemented and the rough scale of parameters it has.

4. Please elaborate on the pre-training dataset of CAML in “we pre-train CAML’s Transformer encoder on few-shot image classification tasks from ImageNet-1k.” How many examples are included in the pre-training set? Note that one of the benchmarks, miniImageNet, is a subset of ImageNet. Would this pretraining dataset result in task leakage?

5. Please elaborate on how the ProtoNet baseline is implemented. Is it trained on the same pre-training dataset but with the ProtoNet loss objective?

---

> ### Author Response · Authors · 2023-11-17
> **Response to Reviewer GdKg**
>
> Thank you for reviewing our work! It’s encouraging that you find our work is well-motivated, presents a compelling analysis with Figure 2, contributes an appropriate theoretical analysis, and presentes strong empirical results on various meta-learning baselines. Addressing your concerns:
>
> >1a. Novelty: the paper does not discuss previous work that also formulated meta-learning as a sequence. Please summarize the novelty of this paper in relation to [1].
>
> Thank you for bringing this to our attention – from your and other reviewers’ feedback, we’ve been able to significantly improve our Related Work. We’ve re-written the related work to include a section on meta-learning as sequence modeling. Based on community feedback, we’ve also included a separate section on in-context learning for other applications of computer vision (i.e. in-painting, depth estimation, etc.).
>
> Addressing your question, [1] is causal: any element in the sequence cannot use later elements to influence its own learned representation. Similar to [1], we initially tried a causal approach; however, we couldn’t get this to work and it also conflicted with our intuition—and the intuition of Set Transformer [2]—that permutation invariance would be an important property for meta-learning applications. This also affects empirical performance. In Section 5.2, we observe an average difference of 5.2 (coincidentally the difference is equal to the section number) accuracy points between CAML and SNAIL [1] and hypothesize the causal approach actually forces a subtly more difficult modeling problem by imposing a causal inductive bias on a fundamentally non-causal prediction task.
>
> **SNAIL Implementation.** We use the architecture presented in [1] but with the hidden dimension of the Attention and Temporal Convolution Blocks adapted to CLIP embeddings rather than the ResNet embeddings used in the original implementation. As in this [1], we freeze the feature extractor and train the SNAIL model parameters during large-scale pre-training.
>
> | Approach | (5w-1s) CIFAR-fs | (5w-1s) MiniImageNet | (5w-1s) Pascal + Paintings | (5w-1s) Paintings | (5w-1s) Pascal |
> | :-----: | :-----: | :-----: | :-----: | :-----: | :-----: |
> |SNAIL| 62.1 | 93.6 | 62.5 | **51.9** | 79.7 |
> | CAML | **70.8** | **96.2** | **63.8** | 51.1 | **82.6** |
>
> | Approach | (5w-1s) meta-iNat | (5w-1s) tiered meta-iNat | (5w-1s) ChestX | (5w-1s) CUB | (5w-1s) tiered-ImageNet | (5w-1s) Aircraft |
> | :-----: | :-----: | :-----: | :-----: | :-----: | :-----: | :-----: |
> |SNAIL | 89.1 | 77.3 | 20.2 | 87.5 | 93.1 | 48.9 |
> | CAML | **91.2** | **81.9** | **21.5** | **91.8** | **95.4** | **63.3**|
>
> | Approach | (5w-5s) CIFAR-fs | (5w-5s) MiniImageNet | (5w-5s) Pascal + Paintings | (5w-5s) Paintings | (5w-5s) Pascal |
> | :-----: | :-----: | :-----: | :-----: | :-----: | :-----: |
> |SNAIL | 71.1 | 98.1 | 77.6 | **65.8** | 88.0 |
> |CAML | **85.5** | **98.6** | **78.3** | 65.2 | **89.7**|
>
> | Approach | (5w-5s) meta-iNat | (5w-5s) tiered meta-iNat | (5w-5s) ChestX | (5w-5s) CUB | (5w-5s) tiered-ImageNet | (5w-5s) Aircraft |
> | :-----: | :-----: | :-----: | :-----: | :-----: | :-----: | :-----: |
> | SNAIL | 94.8 | 86.5 | 20.0 | 92.8 | 97.3 | 35.8 |
> | CAML | **96.3** | **91.6** | **22.2** | **97.1** | **98.1** | **79.1** |
>
> >1b. Please summarize the novelty of this paper in relation to [2].
>
> We share a common insight with [2]: permutation invariance is an important property for many applications of machine learning, and especially meta-learning. However, the contribution of [2] is to provide a permutation invariant sequence-to-sequence architecture, not develop a meta-learning algorithm. Using an analogy, if a meta-learning is a car, then [2] is an engine that might be installed in that car. Indeed [2] says as much in their conclusion, “An interesting future work would be to apply Set Transformer to meta-learning problems.”
>
> In the language of [2], the Transformer encoder of CAML is implemented as SAB, or simply, a self-attention block without positional embeddings. Inspired by your comment, we experimented with switching out SAB for the ISAB block described by [2] during the rebuttal period, but found issues with convergence, likely due to the additional complexity of the ISAB mechanism.

---

> ### Author Response · Authors · 2023-11-17
> **Response to Reviewer GdKg Part II**
>
> >2. Please respond to Weakness 2: The dichotomy between “meta-training“ and ”universal meta-learning.”
>
> Thank you for posing this question! We think this is a critical distinction, and based on your review, it is clear that we failed to communicate this point in the initial manuscript. We specifically address your question in our response to all reviews, but we’d like to extend our response here as well.
>
> *What is the motivation?* We’d like to develop an evaluation paradigm for meta-learning models that quantifies how they would perform in a ChatGPT-like application. Specifically, how do meta-learning algorithms perform when a user can upload any set of images and serving-time constraints prevents fine-tuning on the support set?
>
> *What is missing in current meta-learning evaluation paradigms?* In-domain meta-learning paradigms measure generalization to unseen classes during inference by training on closely related classes during training (i.e. meta-training). Cross-domain meta-learning refers to a challenging evaluation paradigm where the meta-training and inference-time data distributions are significantly different, and the goal is to maximize the performance on the inference-time data distribution by fine-tuning on the support set.
>
> There always seems to be a one-to-one or many-to-one correspondence between the set of classes used for training and the set of classes used during inference. A sense of many-to-many, or simply “do well on everything”, isn’t captured by either in-domain or cross-domain evaluation paradigms, and even for cross-domain settings, the performance on classes similar to those used for meta-training is often not evaluated.
>
> *What does setting universal meta-learning specifically address?* Universal meta-learning specifically addresses a serving-time application of few-shot image classification where the user can upload any set of images and only specify which images are similar (i.e. belong to the same class). Meta-training on a set of related images is impossible, because the space of all possible images that could be uploaded by the user can’t be covered. Moreover, fine-tuning on the support set is also not feasible for inference-time applications deployed in a ChatGPT-like setting.
>
> We propose the universal meta-learning evaluation setting to measure progress in this area. This type of evaluation paradigm already exists for NLP—and is quite popular—but we are unfamiliar with one if it already exists for few-shot image classification.
>
> *Aside.*
>
> If your comment simply refers to loose terminology around “without meta-training”—we use “without meta-training”, “without meta-training on related classes”, and “without meta-training on benchmark training sets” interchangeably—then we’re happy to tighten this up.

---

> ### Author Response · Authors · 2023-11-17
> **Response to Reviewer GdKg Part III**
>
> >3. Please elaborate on how the transformer encoder is implemented and the rough scale of parameters it has.
>
> We use a ViT-Large Transformer Encoder as described in Table 1 of  [3]. It has 302 million trainable parameters. We have added an in-depth description of its implementation in Section B of the Appendix under the header CAML Implementation.
>
> >4. Please elaborate on the pre-training dataset of CAML. Note that one of the benchmarks, miniImageNet, is a subset of ImageNet. Would this pretraining dataset result in task leakage?
>
> All methods evaluated in the *universal meta-learning* setting adhere to the same pre-training paradigm. For each large-scale image classification dataset, we reformulate the objective from typical supervised image classification to both a 5-way-1-shot and a 5-way-5-shot episodic prediction tasks. Within a dataset, examples from different classes are randomly sampled to compose a batch of episodes, and after exhausting iterating through every training example, this process is repeated with the next dataset. Iterating through each dataset in our set of ImageNet-1k, Fungi, MSCOCO, and WikiArt then constitutes a single epoch of training.
>
> We have added this discussion as well as additional information regarding optimization settings to Section B of the Appendix, paragraphs Large-Scale Pre-training and Optimization settings. Please let us know if anything is unclear or missing.
>
> *Task leakage?*
>
> There is task leakage in the mini-ImageNet evaluation; however, this has become standard practice on mini-ImageNet as many feature extractors are pre-trained on ImageNet-1k — please see the excellent discussion of this topic in [4] (specifically, the footnote on page 6 as well as the the paragraph “Class overlap between pre-training and meta-testing”). Tangibly, all baselines in our experimental analysis pre-train on ImageNet-1k (including P>M>F as it uses a DINO feature extractor), so we believe the comparison is fair.
>
> In the context of *universal meta-learning* evaluation, we see mini-ImageNet as a benchmark that evaluates how different meta-learning algorithms perform when evaluated on classes that they have previously seen, similar to how a LLM might recall a memorized answer to a question seen during training.
>
> >5. Please elaborate on how the ProtoNet baseline is implemented. Is it trained on the same pre-training dataset but with the ProtoNet loss objective?
>
> Precisely; it is trained on the same large-scale pre-training dataset but with the ProtoNet loss function. We have added an in-depth discussion of the ProtoNet baseline implementation and training paradigm in Section B of the Appendix under paragraph ProtoNet and MetaOpt Implementations.
>
> ---
>
> [1] Mishra, Nikhil, Mostafa Rohaninejad, Xi Chen, and Pieter Abbeel. "A simple neural attentive meta-learner.", ICLR 2018
>
> [2] Lee, J., Lee, Y., Kim, J., Kosiorek, A., Choi, S. & Teh, Y.W.. (2019). Set Transformer: A Framework for Attention-based Permutation-Invariant Neural Networks., ICML 2019
>
> [3] An Image is Worth 16x16 Words: Transformers for Image Recognition at Scale, Dosovitskiy et al., ICLR 2021
>
> [4] Pushing the Limits of Simple Pipelines for Few-Shot Learning: External Data and Fine-Tuning Make a Difference, Hu et al., CVPR 2022

---

> > ### Author Response · Authors · 2023-11-22
> > **Follow-up**
> >
> > Dear Reviewer GdKg,
> >
> > Thank you again for reviewing our work! As the discussion period ends shortly, we wanted to check if you have any further questions or found our responses helpful?
> >
> > Please let us know and thank you for your time!

---

> > > ### Comment · Reviewer_GdKg · 2023-11-22
> > > **Thanks for the response**
> > >
> > > I'd like to thank the authors for the detailed response, as well as the additional experiments on related methods. Although the overall quality and presentation of the paper have improved, I still think the novelty in relation to [1,2] weakens the contribution of this paper. Furthermore, I do not think the notion of "universal meta-learning" is an appropriate term for the proposed approach; chatgpt-style evaluation paradigm may not be "universal" from the perspective of the meta-learning problem formulation. I'm optimistic about the proposed approach, but I think the authors should better formulate the problem in relation to the meta-learning literature.

---

### Official Review · Reviewer_8Akm · 2023-11-01

**Soundness:** 3 good
**Presentation:** 3 good
**Contribution:** 2 fair
**Rating:** 5
**Confidence:** 5

**Summary:**

This paper addresses a gap in the field of visual meta-learning, where models have traditionally struggled to learn new visual concepts during inference without fine-tuning, a capability that Large Language Models (LLMs) like ChatGPT have demonstrated in the textual domain. The authors introduce a novel meta-learning algorithm inspired by the in-context learning of LLMs. This approach treats n-way-k-shot image classification as a sequence modeling over known labeled data points and an unknown test data point.

**Strengths:**

1. The paper is well-written.
2. The problem studied in this paper is interesting and valuable.
3. The theoretical work of this paper is sufficient, which improves the value of the paper.

**Weaknesses:**

1. The authors utilize the CLIP model to encode both images and labels. An area of potential exploration is why they didn't attempt to encode context and images directly, especially using datasets like MSCOCO.
2. In the experiments, CAML's performance on out-of-domain tasks is notably weak. This might be primarily due to the treatment of unseen categories, which are uniformly encoded as "Unknown [class] Embedding".
3. The study lacks ablation experiments for its various modules, and there's an absence of quantitative analysis for hyperparameters.

**Questions:**

Please see the Weaknesses

---

> ### Author Response · Authors · 2023-11-17
> **Response to Reviewer 8Akm**
>
> Thank you for reviewing our work! We appreciate your assessment that the problem studied in this paper is interesting and valuable as well as the value of our theoretical contribution on label encodings. Addressing your questions:
>
> >1. The authors utilize the CLIP model to encode both images and labels. An area of potential exploration is why they didn't attempt to encode context and images directly, especially using datasets like MSCOCO.
>
> There may be a misunderstanding as our approach does not encode labels with natural language. Specifically, if we are trying to classify a query image from a support set composed of an image of a dog and another image of a cat, then there is no natural language encoding of either “dog” or “cat”. Rather, we only know that the two images in our support set belong to different classes, and our goal is to associate the query image with one of these two classes.
>
> We choose this setting—even when natural language descriptors of an image are available—to evaluate a ChatGPT-like application of few-shot image classification. Specifically, given only a support set containing images and knowledge of which images belong to the same class, can we correctly associate a new image with one of the classes in our support set?
>
> Returning to your question, we can’t directly encode a context and images directly using a dataset like MSCOCO as natural language descriptors are not available during inference. The advantage to this approach is its flexibility: images can be sliced by arbitrary attributes such as the presence of objects or specific textures as visualized in Figure 2.  Another benefit is its simplicity: it involves a single modality (images) and does not need to handle unseen label classes.
>
> >2. In the experiments, CAML's performance on out-of-domain tasks is notably weak. This might be primarily due to the treatment of unseen categories, which are uniformly encoded as "Unknown [class] Embedding".
>
> Building on our previous response, there is no notion of seen or unseen class labels. The model simply knows which images belong to the same class in the support set. We encode the class  information with an ELMES so that image embeddings belonging to the same class are concatenated with the same ELMES label embedding. This allows the model to identify which images in the support set belong to the same (or different) classes.
>
> We have run new experiments to study CAML’s performance on out-of-domain tasks. Specifically, replacing the CLIP image encoder with a ViT-huge pre-trained on Laion-2b increases Aircraft 5-1 accuracy from 63.3 to 81.8 and Aircraft 5-5 accuracy from 79.1 to 92.1. We hypothesize the improvement on Aircraft is due to CLIP embeddings not capturing variability among images, while ViT-h embeddings are better separated. **The analysis we add in Section C of the Appendix under "Image Encoder Ablation" supports this hypothesis and also explains why Laion-2b does not improve CAML’s performance on ChestX.**
>
> | Approach | (5w-1s) CIFAR-fs | (5w-1s) MiniImageNet | (5w-1s) Pascal + Paintings | (5w-1s) Paintings | (5w-1s) Pascal |
> | :-----: | :-----: | :-----: | :-----: | :-----: | :-----: |
> | CAML (CLIP) | 70.8 | 96.2 | 63.8 | 51.1 | 82.6 |
> |CAML (Laion-2b) | **83.3** | **98.6** | **66.4** | **54.7** | **83.4**|
>
> | Approach | (5w-1s) meta-iNat | (5w-1s) tiered meta-iNat | (5w-1s) ChestX | (5w-1s) CUB | (5w-1s) tiered-ImageNet | (5w-1s) Aircraft |
> | :-----: | :-----: | :-----: | :-----: | :-----: | :-----: | :-----: |
> | CAML (CLIP) | 91.2 | 81.9 | **21.5** | 91.8 | 95.4 | 63.3|
> |CAML (Laion-2b) | **94.6** | **89.3**| **21.6** | **95.8** | **96.8**| **81.8**|
>
> | Approach | (5w-5s) CIFAR-fs | (5w-5s) MiniImageNet | (5w-5s) Pascal + Paintings | (5w-5s) Paintings | (5w-5s) Pascal |
> | :-----: | :-----: | :-----: | :-----: | :-----: | :-----: |
> |CAML (CLIP) | 85.5 | 98.6 | 78.3 | 65.2 | 89.7|
> |CAML (Laion-2b) | **93.5** | **99.6** | **81.0** | **69.9** | **90.1** |
>
> | Approach | (5w-5s) meta-iNat | (5w-5s) tiered meta-iNat | (5w-5s) ChestX | (5w-5s) CUB | (5w-5s) tiered-ImageNet | (5w-5s) Aircraft |
> | :-----: | :-----: | :-----: | :-----: | :-----: | :-----: | :-----: |
> | CAML (CLIP) | 96.3 | 91.6 | **22.2** | 97.1 | 98.1 | 79.1 |
> | CAML (Laion-2b) | **97.9** | **95.6** | **22.0** | **98.7** | **98.8** | **92.1**|

---

> > ### Author Response · Authors · 2023-11-17
> > **Response to Reviewer 8Akm Part II**
> >
> > >3. The study lacks ablation experiments for its various modules, and there's an absence of quantitative analysis for hyperparameters.
> >
> > *Lack of hyperparameter quantitative analysis.* This is a great point! Our computational resources are constrained during the rebuttal period, and training a single model takes several days, but we commit to analyzing the robustness of CAML’s performance with respect to different hyperparameter settings for the camera-ready version of the paper.
> >
> > *Lack of Ablation Experiments.* Thank you for leaving this comment – exploring it led us to a really cool analysis and result! There are 3 components of CAML: the image encoder, the label encoder, and the Transformer encoder. We have moved our ablation analysis of the label encoder to the supplementary material due to space constraints in the main text, and **inspired by your comment, we also explore ablating the Image Encoder during the rebuttal period.** Specifically, we expect the performance of CAML to be highly sensitive to the power of the Image Encoder, scaling its performance by the extent to which the Image Encoder captures variability among images in the support and query sets.
> >
> > Our added analysis in Section C indicates this is the case. By switching the Image Encoder from a ViT-base pre-trained with CLIP to a Vit-Huge pre-trained with Laion-2b, the average accuracy across all benchmarks increases by 4.3 points of accuracy. Similarly, replacing the ViT-base Image Encoder pre-trained with CLIP by a ResNet-34 Image Encoder pre-trained on ImageNet-1k, accuracy drops by an average of 6.5 accuracy points. This ablation experiment is added to Section C of the Appendix under header Image Encoder Ablation.

---

> > > ### Author Response · Authors · 2023-11-22
> > > **Follow-up**
> > >
> > > Dear Reviewer 8Akm,
> > >
> > > Thank you again for reviewing our work! As the discussion period ends shortly, we wanted to check if you have any further questions or found our responses helpful?
> > >
> > > Please let us know and thank you for your time!

---

### Author Response · Authors · 2023-11-17
**Meta Response**

We’d like to thank each reviewer for considering our work and their helpful suggestions. We have carefully considered and answered each question. We have also added new experimental results and updated the manuscript—changes highlighted in red—following the suggestions of the reviewers.

## Changes to the manuscript
* Reviewer 8Akm's suggestions.
  * Image encoder albation experimental results (Table 9, Table 10, Table 11, Table 12; Appendix C)
  * Image encoder ablation analysis: why is performance improved on Aircraft but not ChestX? (Figure 4, Appendix C)

* Reviewer GdKg's suggestions.
  * Extension of the Related Work (Section 2)
  * Extension to the Introduction to clarify *universal meta-learning* vs. *meta-training* (Section 1)
  * Empirical comparison with related work SNAIL [1] (Table 1, Table 2, Table 3, Table 4)
  * Addition of an in-depth discussion of experimental settings (Appendix B)

* Reviewer yxC1's suggestions.
  * Addition of an in-depth discussion of experimental settings (Appendix B).
  * Discussion of fine-tuning the image encoder (Appendix D)
  * Clarification of the initialization of the unknown class embedding (Section 3)

* Reviewer Hb5r's suggestions.
  * Empirical comparison with related work GPICL [2] (Table 1, Table 2, Table 3, Table 4)
  * Addition of a section discussing meta-learning label symmetry (Figure 5, Appendix C)
  * Clarification of the problem and our primary contribution (Section 1)

A common theme raised by several reviewers (Hb5r, GdKg) is that our contribution is unclear. Overall, we would like to summarize the problem and our contribution. We also include additional experimental results comparing CAML with GPICL (recommended by Reviewer Hb5r) and SNAIL (recommended by Reviewer GdKg)

## Problem
Visual meta-learning algorithms cannot be deployed to ChatGPT-like systems because such systems require models that can *(1) generalize to a broad set of tasks unknown at training time* and *(2) do so in real-time, without the time allowance for parameter updates* – in other words, they must be "general and fast”. Large Language Models have shown a remarkable ability to do both; however, current visual meta-learners can only satisfy one requirement or the other. Specifically, at inference time, most visual meta-learners can only generalize to tasks similar to those seen during meta-training (“fast but not general”). On the other hand, many works explore generalization to out-of-distribution tasks, but they require fine-tuning the model on the support set of each downstream task, which would be infeasible in a real-time application (“general but not fast”).

## Contribution
Our contribution is twofold.

**1. We develop a new meta-learning evaluation paradigm (*universal meta-learning*) that approximates the performance of visual meta-learning algorithms in a ChatGPT-like application.** That is, we want to see how visual meta-learners perform on diverse tasks unseen at training, with no parameter updates. This is hard: we want a model that will work well for any set of images, but actually deploying to users and quantifying failure cases is impractical for many research groups. Our best proxy to measure a model’s capacity to generalize in this setting is to evaluate it on a diverse set of meta-learning benchmarks spanning many different image classification paradigms without meta-training on any of their training sets or fine-tuning on the support set during inference.

---

> ### Author Response · Authors · 2023-11-17
> **Meta Response Part II**
>
> **2. We develop a meta-learning algorithm that works well in this setting (i.e. CAML).**
> CAML is a sequence-based meta-learning algorithm that is designed to generalize to any set of images during inference without fine-tuning (thereby addressing the problem). In our experiments, we find that CAML surpasses the performance of other meta-learning algorithms in the universal meta-learning evaluation setting. Surprisingly, its performance in the universal meta-learning setting often matches–and even exceeds—the in-domain meta-learning performance of the state-of-the-art meta-learning algorithm, P>M>F, that is directly meta-trained on each benchmark.
>
> CAML is designed to perform well in the universal meta-learning setting – that is, it’s “general and fast” – while prior approaches are not. Specifically, and distinct from prior sequence-based meta-learning algorithms like SNAIL [1] and GPICL [2],
> 1. CAML is invariant to permutations in the support set and non-causal. In our experiments, we observe a performance gap between SNAIL/GPICL approaches and CAML and hypothesize the causal approach actually forces a subtly more difficult modeling problem by imposing a causality inductive bias on a fundamentally non-causal prediction task.
> 2. CAML uses an Equal Length and Maximally Equiangular Set (ELMES) to integrate the label information from the support set into the model. Our theoretical analysis shows this formulation minimizes the entropy of identifying class labels.
> 3. CAML represents an (image, label) element of the support set as a single vector that contains information about both the image and the label. Specifically, a Foundational Model Image Encoder embeds the support set image, an ELMES embeds the label, and both vectors are concatenated to form a single demonstration vector from the (image, label) pair in the support set.
>
> These changes make a big difference. In our experiments, we observe a performance gap of (on average) **5.2 accuracy points between CAML and SNAIL** and **7.3 accuracy points between CAML and GPICL** for each benchmark.
>
> | Approach | (5w-1s) CIFAR-fs | (5w-1s) MiniImageNet | (5w-1s) Pascal + Paintings | (5w-1s) Paintings | (5w-1s) Pascal |
> | :-----: | :-----: | :-----: | :-----: | :-----: | :-----: |
> |GPICL| 41.5 | 95.6 | 62.6 | 51.6 | 81.7 |
> |SNAIL| 62.1 | 93.6 | 62.5 | **51.9** | 79.7 |
> | CAML | **70.8** | **96.2** | **63.8** | 51.1 | **82.6** |
>
> | Approach | (5w-1s) meta-iNat | (5w-1s) tiered meta-iNat | (5w-1s) ChestX | (5w-1s) CUB | (5w-1s) tiered-ImageNet | (5w-1s) Aircraft |
> | :-----: | :-----: | :-----: | :-----: | :-----: | :-----: | :-----: |
> |GPICL | 90.0 | 60.8 | 20.1 | 75.1 | 94.6 | 19.8 |
> |SNAIL | 89.1 | 77.3 | 20.2 | 87.5 | 93.1 | 48.9 |
> | CAML | **91.2** | **81.9** | **21.5** | **91.8** | **95.4** | **63.3**|
>
> | Approach | (5w-5s) CIFAR-fs | (5w-5s) MiniImageNet | (5w-5s) Pascal + Paintings | (5w-5s) Paintings | (5w-5s) Pascal |
> | :-----: | :-----: | :-----: | :-----: | :-----: | :-----: |
> |GPICL | 78.3 | 98.2 | 74.6 | 61.0 | 88.2 |
> |SNAIL | 71.1 | 98.1 | 77.6 | **65.8** | 88.0 |
> |CAML | **85.5** | **98.6** | **78.3** | 65.2 | **89.7**|
>
> | Approach | (5w-5s) meta-iNat | (5w-5s) tiered meta-iNat | (5w-5s) ChestX | (5w-5s) CUB | (5w-5s) tiered-ImageNet | (5w-5s) Aircraft |
> | :-----: | :-----: | :-----: | :-----: | :-----: | :-----: | :-----: |
> |GPICL | 95.1 | 87.6 | 20.9 | 94.5 | 97.2 | 61.8 |
> | SNAIL | 94.8 | 86.5 | 20.0 | 92.8 | 97.3 | 35.8 |
> | CAML | **96.3** | **91.6** | **22.2** | **97.1** | **98.1** | **79.1** |

---

> ### Author Response · Authors · 2023-11-17
> **Meta Response Part III**
>
> We also update the manuscript to include an Image Encoder ablation (recommended by 8Akm) and find that CAML’s performance is increased by (on average) 4.3 points of accuracy by choosing a more powerful Image Encoder.
>
> ## Image Encoder Ablation.
>
> Inspired by the comment from Reviewer 8Akm, we explore Image Encoders other than CLIP. Replacing CLIP with a ViT-huge pretrained on Laion-2b, **CAML’s performance improves by (on average) 4.3% accuracy for each benchmark.** With this change, CAML suprasses the current state-of-the-art meta-learning algorithm that is evaluated on the in-domain setting (i.e., by meta-training on the train set of each benchmark) on all benchmarks except CIFAR-fs 5-5 and ChestX 5-1 and 5-5. We include an analysis of this ablation in Section C of the appendix under paragraph **Image Encoder Ablation.**
>
> | Approach | (5w-1s) CIFAR-fs | (5w-1s) MiniImageNet | (5w-1s) Pascal + Paintings | (5w-1s) Paintings | (5w-1s) Pascal |
> | :-----: | :-----: | :-----: | :-----: | :-----: | :-----: |
> | CAML (CLIP) | 70.8 | 96.2 | 63.8 | 51.1 | 82.6 |
> |CAML (Laion-2b) | **83.3** | **98.6** | **66.4** | **54.7** | **83.4**|
>
> | Approach | (5w-1s) meta-iNat | (5w-1s) tiered meta-iNat | (5w-1s) ChestX | (5w-1s) CUB | (5w-1s) tiered-ImageNet | (5w-1s) Aircraft |
> | :-----: | :-----: | :-----: | :-----: | :-----: | :-----: | :-----: |
> | CAML (CLIP) | 91.2 | 81.9 | **21.5** | 91.8 | 95.4 | 63.3|
> |CAML (Laion-2b) | **94.6** | **89.3**| **21.6** | **95.8** | **96.8**| **81.8**|
>
> | Approach | (5w-5s) CIFAR-fs | (5w-5s) MiniImageNet | (5w-5s) Pascal + Paintings | (5w-5s) Paintings | (5w-5s) Pascal |
> | :-----: | :-----: | :-----: | :-----: | :-----: | :-----: |
> |CAML (CLIP) | 85.5 | 98.6 | 78.3 | 65.2 | 89.7|
> |CAML (Laion-2b) | **93.5** | **99.6** | **81.0** | **69.9** | **90.1** |
>
> | Approach | (5w-5s) meta-iNat | (5w-5s) tiered meta-iNat | (5w-5s) ChestX | (5w-5s) CUB | (5w-5s) tiered-ImageNet | (5w-5s) Aircraft |
> | :-----: | :-----: | :-----: | :-----: | :-----: | :-----: | :-----: |
> | CAML (CLIP) | 96.3 | 91.6 | **22.2** | 97.1 | 98.1 | 79.1 |
> | CAML (Laion-2b) | **97.9** | **95.6** | **22.0** | **98.7** | **98.8** | **92.1**|
>
> Finally, we clarify the experimental settings (recommended by yxC1 and Hb5r).
>
> ## Clarifying Experimental Settings.
>
> We sincerely thank reviewers yxC1 and Hb5r for seeking clarity on our experimental settings. We missed providing this information in the initial draft and have rectified it in Section B in the Appendix of the updated manuscript.
>
> ---
>
> [1] Mishra, Nikhil, Mostafa Rohaninejad, Xi Chen, and Pieter Abbeel. "A simple neural attentive meta-learner.", ICLR 2018
>
> [2] General-Purpose In-Context Learning by Meta-Learning Transformers, NeurIPs 2022

---

### Comment · Area_Chair_9n1L · 2023-12-05
**Final Update**

Dear Reviewers,

Please take this chance to carefully read the rebuttal from the authors and make any final changes if necessary.

Please also respond to the authors that you have read their rebuttal, and give feedback whether their rebuttal have addressed your concerns.

Thank you,

AC

---

### Meta-Review · Area_Chair_9n1L · 2023-12-11

**Metareview:**

In this paper, the authors propose to incorporate an in-context learning training objective in meta learning similar to in-context learning in LLM, using a Transformer encoder + ELMES class encoder design that preserves the label symmetry. The authors also consider a new meta-learning evaluation setting (universal meta-learning). The proposed method improves over a variety of related meta-learning approaches under the same setting.

On the strength side, the paper is overall well-written. Some parts of the paper were hard to follow in the beginning, but were significantly improved following the reviewers' suggestions. The motivation and technical contributions are clear. Literature survey of the revised version is thorough and the latest status of the field is well-presented. The concerns from the reviewers have been sufficiently addressed, including the novelty compared to existing works.

On the weakness side, I find it a bit misleading that the authors keep emphasizing the motivation of the proposed universal meta-learning framework as "ChatGPT-like". While there are indeed connections, this is an over-simplification that may mistakenly exaggerate the scope of this work and create unnecessary confusion/hypes. It is suggested that the authors tone down this part and only briefly mention when introducing the motivation of in-context learning. In addition, ablation studies and analysis of the proposed method fall short despite the ones in the supplementary. Finally, the authors may have a bit overly shown the fundamentalism on using meta-learning to address visual in-context learning. While this is an interesting direction, visual in-context learning is also related to open-set metric learning broadly speaking, which has been already widely explored in both face verification and retrieval. More discussions outside meta-learning should also happen.

Despite the weaknesses, the AC still considers the proposed setting an interesting one that could benefit the meta-learning community. The experiments are sufficiently comprehensive and compelling. For this, the AC recommends acceptance to ICLR.

**Justification For Why Not Higher Score:**

The findings and novelty of this work are not ground breaking given the existing works. In addition, the AC considers that the scope of the problem being addressed and its application somewhat limited.

**Justification For Why Not Lower Score:**

While the scores are a bit borderline, the paper is solid and the quality is above the ICLR threshold. The concerns have been well-addressed.

---

### Decision · Program_Chairs · 2024-01-16

Accept (poster)